# Sparsity Winning Twice: Better Robust Generalization from More Efficient Training

**Tianlong Chen[1*], Zhenyu Zhang[2*], Pengjun Wang[2*], Santosh Balachandra[1*],**
**Haoyu Ma[3*], Zehao Wang[2], Zhangyang Wang[1]**
[1]University of Texas at Austin, [2]University of Science and Technology of China,
[3]University of California, Irvine
{tianlong.chen,santoshb,atlaswang}@utexas.edu,
{zzy19969,wpj520,wangze}@mail.ustc.edu.cn, haoyum3@uci.edu

## Abstract

Recent studies demonstrate that deep networks, even robustified by the state-of-the-art adversarial training (AT), still suffer from large robust generalization gaps, in addition to the much more expensive training costs than standard training. In this paper, we investigate this intriguing problem from a new perspective, i.e., *injecting appropriate forms of sparsity* during adversarial training. We introduce two alternatives for sparse adversarial training: (i) *static sparsity*, by leveraging recent results from the lottery ticket hypothesis to identify critical sparse subnetworks arising from the early training; (ii) *dynamic sparsity*, by allowing the sparse subnetwork to adaptively adjust its connectivity pattern (while sticking to the same sparsity ratio) throughout training. We find both static and dynamic sparse methods to yield win-win: substantially shrinking the robust generalization gap and alleviating the robust overfitting, meanwhile significantly saving training and inference FLOPs. Extensive experiments validate our proposals with multiple network architectures on diverse datasets, including CIFAR-10/100 and Tiny-ImageNet. For example, our methods reduce robust generalization gap and overfitting by $34.44\%$ and $4.02\%$, with comparable robust/standard accuracy boosts and $87.83\%/87.82\%$ training/inference FLOPs savings on CIFAR-100 with ResNet-18. Besides, our approaches can be organically combined with existing regularizers, establishing new state-of-the-art results in AT. Codes are available in `https://github.com/VITA-Group/Sparsity-Win-Robust-Generalization`.

## 1 Introduction

Deep neural networks (DNNs) are notoriously vulnerable to maliciously crafted adversarial attacks. To conquer this fragility, numerous adversarial defense mechanisms are proposed to establish robust neural networks (Schmidt et al., 2018; Sun et al., 2019; Nakkiran, 2019; Raghunathan et al., 2019; Hu et al., 2019; Chen et al., 2020c; 2021e; Jiang et al., 2020). Among them, *adversarial training* (AT) based methods (Madry et al., 2017; Zhang et al., 2019) have maintained the state-of-the-art robustness. However, the AT training process usually comes with order-of-magnitude higher computational costs than standard training, since multiple attack iterations are needed to construct strong adversarial examples (Madry et al., 2018b). Moreover, AT was recently revealed to incur severe robust generalization gaps (Rice et al., 2020), between its training and testing accuracies, as shown in Figure 1; and to require significantly more training samples (Schmidt et al., 2018) to generalize robustly.

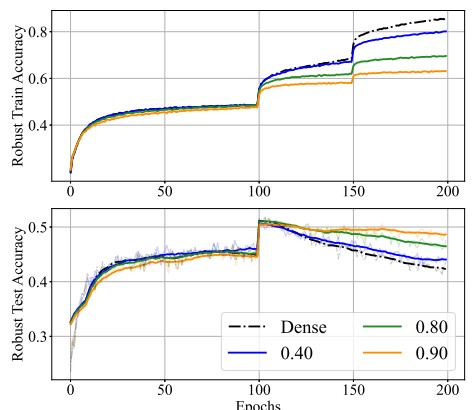

Figure 1: Robust train / test accuracy (*Top / Bottom*) on CIFAR-10 with ResNet-18 across various sparsity levels from $0\%$ (Dense) to $90\%$. The dash-dot and solid lines represent the vanilla PGD-AT dense baseline and our sparse proposals, respectively. As the sparsity increases, the robust generalization gap between training and testing accuracy is substantially narrowed.

---

*Equal Contribution.

In response to those challenges, Schmidt et al. (2018); Lee et al. (2020); Song et al. (2019) investigate the possibility of improving generalization by leveraging advanced data augmentation techniques, which further amplifies the training cost of AT. Recent studies (Rice et al., 2020; Chen et al., 2021e) found that early stopping, or several smoothness/flatness-aware regularizations (Chen et al., 2021e; Stutz et al., 2021; Singla et al., 2021), can bring effective mitigation.

In this paper, a new perspective has been explored to tackle the above challenges by *enforcing appropriate sparsity patterns* during AT. The connection between robust generalization and sparsity is mainly inspired by two facts. On one hand, sparsity can effectively regularize the learning of over-parameterized neural networks, hence potentially benefiting both standard and robust generalization (Balda et al., 2019). As demonstrated in Figure 1, with the increase of sparsity levels, the robust generalization gap is indeed substantially shrunk while the robust overfitting is alleviated. On the other hand, one key design philosophy that facilitates this consideration is the lottery ticket hypothesis (LTH) (Frankle & Carbin, 2019). The LTH advocates the existence of highly sparse and separately trainable subnetworks (a.k.a. winning tickets), which can be trained from the original initialization to match or even surpass the corresponding dense networks' test accuracies. These facts point out a promising direction that utilizing proper sparsity is capable of boosting robust generalization while maintaining competitive standard and robust accuracy.

Although sparsity is beneficial, the current methods (Frankle & Carbin, 2019; Frankle et al., 2020; Renda et al., 2020) often empirically locate sparse critical subnetworks by Iterative Magnitude Pruning (IMP). It demands excessive computational cost even for standard training due to the iterative train-prune-retrain process. Recently, You et al. (2020) demonstrated that these intriguing subnetworks can be identified at the very early training stage using one-shot pruning, which they term as *Early Bird* (EB) tickets. We show the phenomenon also exists in the adversarial training scheme. More importantly, we take one leap further to reveal that even in adversarial training, EB tickets can be drawn from a cheap standard training stage, while still achieving solid robustness. In other words, *the Early Bird is also a Robust Bird* that yields an attractive win-win of efficiency and robustness - we name this finding as *Robust Bird* (RB) tickets.

Furthermore, we investigate the role of sparsity in a scene where the sparse connections of subnetworks change on the fly. Specifically, we initialize a subnetwork with random sparse connectivity and then optimize its weights and sparse typologies simultaneously, while sticking to the fixed small parameter budget. This training pipeline, called as *Flying Bird* (FB), is motivated by the latest sparse training approaches (Evci et al., 2020b) to further reduce robust generalization gap in AT, while ensuring low training costs. Moreover, an enhanced algorithm, i.e., *Flying Bird+*, is proposed to dynamically adjust the network capacity (or sparsity) to pursue superior robust generalization, at few extra prices of training efficiency. Our contributions can be summarized as follows:

- We perform a thorough investigation to reveal that introducing appropriate sparsity into AT is an appealing win-win, specifically: (1) substantially alleviating the robust generalization gap; (2) maintaining comparable or even better standard/robust accuracies; and (3) enhancing the AT efficiency by training only compact subnetworks.

- We explore two alternatives for sparse adversarial training: (i) the *Robust Bird* (RB) training that leverages static sparsity, by mining the critical sparse subnetwork at the early training stage, and using only the cheapest standard training; (ii) the *Flying Bird* (FB) training that allows for dynamic sparsity, which jointly optimizes both network weights and their sparse connectivity during AT, while sticking to the same sparsity level. We also discuss a FB variant called *Flying Bird+* that adaptively adjusts the sparsity level on demand during AT.

- Extensive experiments are conducted on CIFAR-10, CIFAR-100, and Tiny-ImageNet with diverse network architectures. Specifically, our proposals obtain $80.16\% \sim 87.83\%$ training FLOPs and $80.16\% \sim 87.83\%$ inference FLOPs savings, shrink robust generalization from $28.00\% \sim 63.18\%$ to $4.43\% \sim 34.44\%$, and boost the robust accuracy by up to $0.60\%$ and the standard accuracy by up to $0.90\%$, across multiple datasets and architectures. Meanwhile, combining our sparse adversarial training frameworks with existing regularizations establishes the new state-of-the-art results.

## 2 RELATED WORK

**Adversarial training and robust generalization/overfitting.** Deep neural networks present vulnerability to imperceivable adversarial perturbations. To deal with this drawback, numerous defense

approaches have been proposed (Goodfellow et al., 2015; Kurakin et al., 2016; Madry et al., 2018a). Although many methods (Liao et al., 2018; Guo et al., 2018a; Xu et al., 2017; Dziugaite et al., 2016; Dhillon et al., 2018a; Xie et al., 2018; Jiang et al., 2020) were later found to result from obfuscated gradients (Athalye et al., 2018), adversarial training (AT) (Madry et al., 2018a), together with some of its variants (Zhang et al., 2019; Mosbach et al., 2018; Dong et al., 2018), remains as one of the most effective yet costly approaches.

A pitfall of AT, i.e., the poor robust generalization, was spotted recently. Schmidt et al. (2018) showed that AT intrinsically demands a larger sample complexity to identify well-generalizable robust solutions. Therefore, data augmentation (Lee et al., 2020; Song et al., 2019) is an effective remedy. Stutz et al. (2021); Singla et al. (2021) related robust generalization gap to curvature/flatness of loss landscapes. They introduced weight perturbing approaches and smooth activation functions to reshape the loss geometry and boost robust generalization ability. Meanwhile, the robust overfitting (Rice et al., 2020) in AT usually happens with or as a result of inferior generalization. Previous studies (Rice et al., 2020; Chen et al., 2021e) demonstrated that conventional regularization-based methods (e.g., weight decay and simple data augmentation) can not alleviate robust overfitting. Then, numerous advanced algorithms (Zhang et al., 2020; 2021b; Zhou et al., 2021; Bunk et al., 2021; Chen et al., 2021a; Dong et al., 2021; Zi et al., 2021; Tack et al., 2021; Zhang et al., 2021a) arose in the last half year to tackle the overfitting, using data manipulation, smoothened training, and else. Those methods work orthogonally to our proposal as evidenced in Section 4.

Another group of related literature lies in the field of sparse robust networks (Guo et al., 2018b). These works either treat model compression as a defense mechanism (Wang et al., 2018; Gao et al., 2017; Dhillon et al., 2018b) or pursue robust and efficient sub-models that can be deployed in resource-limited platforms (Gui et al., 2019; Ye et al., 2019; Sehwag et al., 2019). Compared to those inference-focused methods, our goal is fundamentally different: injecting sparsity during training to reduce the robust generalization gap while improving training efficiency.

**Static pruning and dynamic sparse training.** Pruning (LeCun et al., 1990; Han et al., 2015a) serves as a powerful technique to eliminate the weight redundancy in over-parameterized DNNs, which aims to obtain storage and computational savings with almost undamaged performance. It can roughly divided into two categories based on how to generate sparse patterns: ($i$) *static pruning*. It removes parameters (Han et al., 2015a; LeCun et al., 1990; Han et al., 2015b) or substructures (Liu et al., 2017; Zhou et al., 2016; He et al., 2017) based on optimized importance scores (Zhang et al., 2018; He et al., 2017) or some heuristics like weight magnitude (Han et al., 2015a), gradient (Molchanov et al., 2019), hessian (LeCun et al., 1990) statistics. The discarded elements usually will not participate in the next round of training or pruning. Static pruning can be flexibly applied prior to training, such as SNIP (Lee et al., 2019), GraSP (Wang et al., 2020) and SynFlow (Tanaka et al., 2020); during training (Zhang et al., 2018; He et al., 2017); and post training (Han et al., 2015a) for different trade-off between training cost and pruned models' quality. ($ii$) *dynamic sparse training*. It updates model parameters and sparse connectivities at the same time, starting from a randomly sparsified subnetwork (Molchanov et al., 2017). During the training, the removed elements have chances to be grown back if they potentially benefit to predictions. Among the huge family of sparse training (Mocanu et al., 2016; Evci et al., 2019; Mostafa & Wang, 2019; Liu et al., 2021a; Dettmers & Zettlemoyer, 2019; Jayakumar et al., 2021; Raihan & Aamodt, 2020), the recent methods Evci et al. (2020a); Liu et al. (2021b) lead to the state-of-the-art performance.

A special case of static pruning, Lottery tickets hypothesis (LTH) (Frankle & Carbin, 2019), demonstrates the existence of sparse subnetworks in DNNs, which are capable of training in isolation and reach a comparable performance of their dense counterpart. The LTH indicates the great potential to train a sparse network from scratch without sacrificing expressiveness and has recently drawn lots of attention from diverse fields (Chen et al., 2020b;a; 2021g;f;d;c;b; 2022; Ding et al., 2022; Gan et al., 2021) beyond image recognition (Zhang et al., 2021d; Frankle et al., 2020; Redman et al., 2021).

## 3 METHODOLOGY

### 3.1 PRELIMINARIES

**Adversarial training (AT).** As one of the widely adopted defense mechanisms, adversarial training (Madry et al., 2018b) effectively tackles the vulnerability to maliciously crafted adversarial samples. As formulated in Equation 1, AT (specifically PGD-AT) replaces the original empirical risk minimization into a min-max optimization problem:

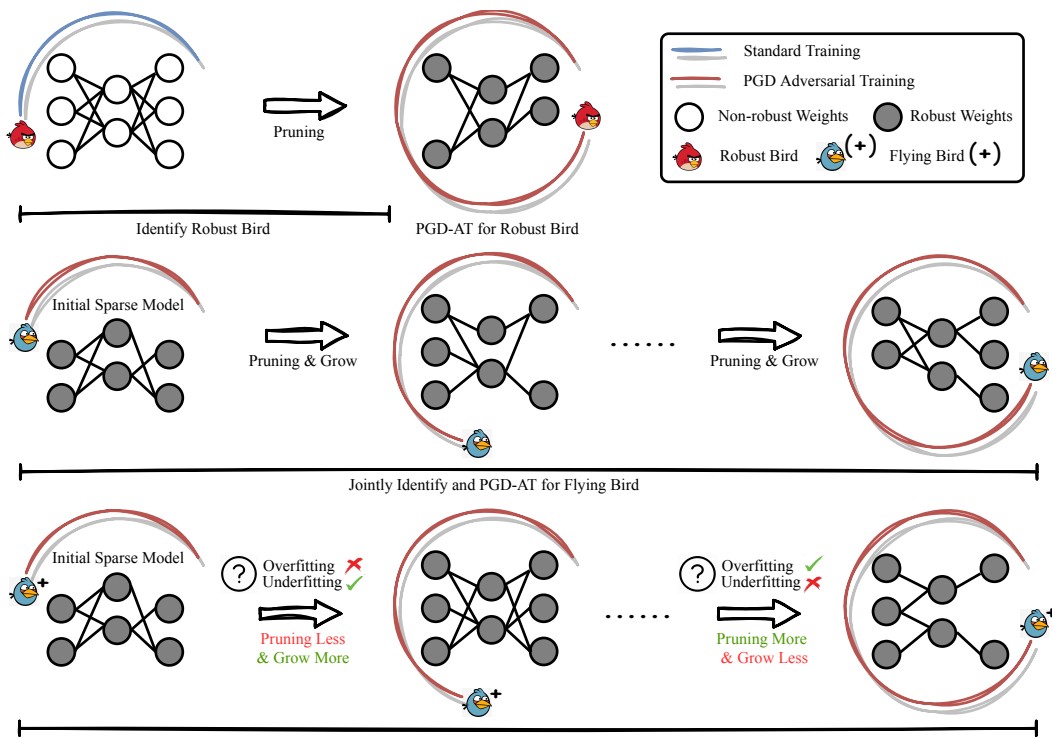

Figure 2: Overview of our proposed training frameworks including Robust Bird (RB), Flying Bird (FB), and Flying Bird (FB+). The length of cycles roughly indicates the number of training epochs.

$$\min_{\theta} \mathbb{E}_{(x,y)\in\mathcal{D}} \mathcal{L}\big(f(x;\theta),y\big) \implies \min_{\theta} \mathbb{E}_{(x,y)\in\mathcal{D}} \max_{\|\delta\|_p \leq \epsilon} \mathcal{L}\big(f(x+\delta;\theta),y\big), \tag{1}$$

where $f(x;\theta)$ is a network with parameters $\theta$. Input data $x$ and its associated label $y$ from training set $\mathcal{D}$ are used to first generate adversarial perturbations $\delta$ and then minimize the empirical classification loss $\mathcal{L}$. To meet the imperceptible requirement, the $\ell_p$ norm of $\delta$ is constrained by a small constant $\epsilon$. Projected Gradient Descent (PGD), i.e., $\delta^{t+1} = \text{proj}_{\mathcal{P}}[\delta^t + \alpha \cdot \text{sgn}\big(\nabla_x\mathcal{L}\big(f(x+\delta^t;\theta),y\big)\big)]$, is usually utilized to produce the adversarial perturbations with step size $\alpha$, which works in an iterative manner leveraging the local first order information about the network (Madry et al., 2018b).

**Sparse subnetworks.** Following the routine notations in Frankle & Carbin (2019), $f(x; m \odot \theta)$ donates a sparse subnetwork with a binary pruning mask $m \in \{0,1\}^{\|\theta\|_0}$, where $\odot$ is the element-wise product. Intuitively, it is a copy of dense network $f(x;\theta)$ with a portion of fixed *zero* weights.

## 3.2 ROBUST BIRD FOR ADVERSARIAL TRAINING

**Introducing Robust Bird.** The primary goal of *Robust Bird* is to find a high-quality sparse subnetwork efficiently. As shown in Figure 2, it locates subnetworks *quickly* by detecting critical network structures arising in the early training, which later can be robustified with much less computation.

Specifically, for each epoch $t$ during training, *Robust Bird* creates a sparsity mask $m_t$ by "masking out" the $p\%$ lowest-magnitude weights; then, *Robust Bird* tracks the corresponding mask dynamics. The **key observation** behind *Robust Bird* is that the sparsity mask $m_t$ does *not* change drastically beyond the early epochs of training (You et al., 2020) because high-level network connectivity patterns are learned during the initial stages (Achille et al., 2019). This indicates that *(i)* winning tickets emerge at a very early training stage, and *(ii)* that they can be identified efficiently.

*Robust Bird* exploits this observation by comparing the Hamming distance between sparsity masks found in consecutive epochs. For each epoch, the last $l$ sparsity masks are stored. If all the stored masks are sufficiently close to each other, then the sparsity masks are not changing drastically over time and network connectivity patterns have emerged; thus, a *Robust Bird* ticket (RB ticket) is drawn. A detailed algorithmic implementation is provided in Algorithm 1 of Appendix A1. This is the RB ticket used in the second stage of adversarial training.

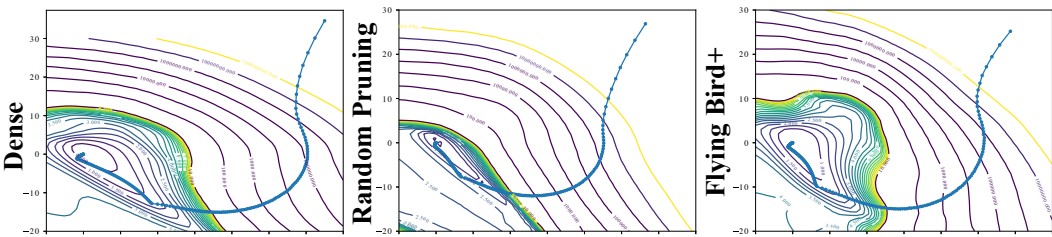

Figure 3: Visualization of loss contours and training trajectories. We compare the dense network, randomly pruned sparse networks, and *flying bird+* at 90% sparsity from ResNet-18 robustified on CIFAR-10.

**Rationale of Robust Bird.** Recent studies (Zhang et al., 2021c) present theoretical analyses that identified sparse winning tickets enlarge the convex region near the good local minima, leading to improved generalization. Our work also shows a related investigation in Figure A9 that, compared with dense models and random pruned subnetworks, RB tickets found by the standard training have much flatter loss landscapes, serving a high-quality **starting point** for further robustification. This occurs because *flatness* of the loss surface is often believed to indicate the standard generalization. Similarly, as advocated by Wu et al. (2020a); Hein & Andriushchenko (2017), a flatter adversarial loss landscape also effectively shrinks the robustness generalization gap. This "flatness preference" of adversarial robustness has been revealed by numerous empirical defense mechanisms, including Hessian/curvature-based regularization (Moosavi-Dezfooli et al., 2019), learned weight and logits smoothening (Chen et al., 2021e), gradient magnitude penalty (Wang & Zhang, 2019), smoothening with random noise (Liu et al., 2018), or entropy regularization (Jagatap et al., 2020).

These observations make the main cornerstone for our proposal and provide possible interpretations to the surprising finding that the RB tickets pruned from a *non-robust* model can be used for obtaining well-generalizable robust models in the followed robustification. Furthermore, unlike previous costly flatness regularizers (Moosavi-Dezfooli et al., 2019), our methods not only offer a flatter starting point but also obtain substantial computational savings due to the reduced model size.

### 3.3 FLYING BIRD FOR ADVERSARIAL TRAINING

**Introducing Flying Bird(+).** Since sparse subnetworks from static pruning are unable to regret for removed elements, they may be too aggressive to capture the pivotal structural patterns. Thus, we introduce *Flying Bird* (FB) to conduct a thorough exploration of dynamic sparsity, which allows pruned parameters to be grown back and engages in the next round of training or pruning, as demonstrated in Figure 2. Specifically, it starts from a sparse subnetwork $f(x; m \odot \theta)$ with a random binary mask $m$, and then jointly optimize model parameters and sparse connectivities simultaneously. In other words, the subnetwork's typologies are "on the fly", decided dynamically based on current training status. Specifically, we update *Flying Bird*'s sparse connectivity every $\Delta t$ epochs of adversarial training, which consists of two continually applied operations: pruning and growing. For the pruning step, $p\%$ of model weights with the lowest magnitude will be eliminated, while $g\%$ weights with the largest gradient will be added back in the growth step. Note that newly added connections are not activated in the last sparse topology, and are initialized to zero since it establishes better performance as indicated in (Evci et al., 2020a; Liu et al., 2021b). *Flying Bird* maintains the sparsity ratio unchanged during the full training by keeping both pruning and growing ratio $p\%, g\%$ equal $k\%$ that decays with a cosine annealing schedule.

We further propose *Flying Bird+*, an enhanced variant of FB, capable of adaptively adjusting the sparsity and learning the right parameterization level "on demand" during training, as shown in Figure 2. To be specific, we first record the robust generalization gap and robust validation loss at each training epoch. An increasing generalization gap of the later training stage indicates a risk of overfitting, while a plateau validation loss implies underfitting. Hence, we then analyze the fitting status according to the upward/downward trend of those measurements. If most epochs (e.g., more than 3 out of the past 5 epochs in our case) tend to see enlarged robust generalization gaps, we raise the pruning ratio $p\%$ to further trim down the network capacity. Similarly, if the majority of epochs present unchanged validation loss, we will increase the growing ratio $q\%$ to enrich the subnetwork capacity. Detailed procedures are summarized in Algorithm 2 of Appendix A1.

**Rationale of Flying Bird(+).** As demonstrated in Evci et al. (2020a), allowing new connections to grow yields improved flexibility in navigating the loss surfaces, which creates the opportunity to

escape bad local minima and search for the optimal sparse connectivity Liu et al. (2021b). *Flying Bird* follows a similar design philosophy that excludes least important connections (Han et al., 2015a) while activating new connections with the highest potential to decrease the training loss fastest. Recent works (Wu et al., 2020c; Liu et al., 2019) have also found enabling network (re-)growth can turn a poor local minima into a saddle point that facilitates further loss decrease. *Flying Bird+* empowers the flexibility further by adaptive sparsity level control.

The flatness of loss geometry provides another view to dissect the robust generalization gain (Chen et al., 2021e; Stutz et al., 2021; Singla et al., 2021). Figure 3 compares the loss landscapes and training trajectories of dense, randomly pruned subnetworks, and Flying Brid+ robustified on CIFAR-10. We observe that *Flying Bird+* converges to a wider loss valley with improved flatness, which usually suggests superior robust generalization (Wu et al., 2020a; Hein & Andriushchenko, 2017). Last but not the least, our approaches also significantly trim down both the training memory overhead and the computational complexity, enjoying extra bonus of efficient training and inference.

## 4 EXPERIMENT RESULTS

**Datasets and architectures.** Our experiments consider two popular architectures, ResNet-18 (He et al., 2016), VGG-16 (Simonyan & Zisserman, 2014) on three representative datasets, CIFAR-10, CIFAR-100 (Krizhevsky & Hinton, 2009) and Tiny-ImageNet (Deng et al., 2009). We randomly split one-tenth of the training samples as the validation dataset, and the performance is reported on the official testing dataset.

**Training and evaluation details.** We implement our experiments with the original PGD-based adversarial trainig (Madry et al., 2018b), in which we train the network against $\ell_\infty$ adversary with maximum perturbations $\epsilon$ of $8/255$. 10-steps PGD for training and 20-steps PGD for evaluation are chosen with a step size $\alpha$ of $2/255$, following Madry et al. (2018b); Chen et al. (2021e). In addition, we also use Auto-Attack (Croce & Hein, 2020) and CW Attack (Carlini & Wagner, 2017) for a more rigorous evaluation. More details are provided in Appendix A2. For each experiment, we train the network for 200 epochs with an SGD optimizer, whose momentum and weight decay are kept to 0.9 and $5 \times 10^{-4}$, respectively. The learning rate starts from 0.1 that decays by 10 times at 100,150 epoch and the batch size is 128, which follows Rice et al. (2020).

For *Robust Bird*, the threshold $\tau$ of mask distance is set as 0.1. In *Flying Birds*(+), we calculate the layer-wise sparsity by Ideal Gas Quotas (IGQ) (Vysogorets & Kempe, 2021) and then apply random pruning to initialize the sparse masks. FB updates the sparse connectivity per 2000 iterations of AT, with an update ratio $k$ that starts from $50\%$ and decays by cosine annealing. More details are referred to Appendix A2. Hyperparameters are either tuned by grid search or following Liu et al. (2021b).

**Evaluation metrics.** In general, we care about both the accuracy and efficiency of obtained sparse networks. To assess the accuracy, we consider both Robust Testing Accuracy (**RA**) and Standard Testing Accuracy (**SA**) which are computed on the perturbed and the original test sets, together with Robust Generalization Gap (**RGG**) (i.e., the gap of RA between train and test sets). Meantime, we report the floating point operations (**FLOPs**) of the whole training process and single image inference to measure the efficiency.

### 4.1 ROBUST BIRD IS A GOOD BIRD

In this section, we evaluate the effectiveness of static sparsity from diverse representative pruning approaches, including: (i) *Random Pruning* (RP), by randomly eliminating model parameters to the desired sparsity; (ii) *One-shot Magnitude Pruning* (OMP), which globally removes a certain ratio of lowest-magnitude weights; (iii) *Pruning at Initialization* algorithms. Three advanced methods, i.e., SNIP (Lee et al., 2019), GraSP (Wang et al., 2020) and SynFlow (Tanaka et al., 2020), are considered, which identify the subnetworks at initialization respect to certain criterion of gradient flow. (iv) *Ideal Gas Quotas* (IGS) (Vysogorets & Kempe, 2021). It adopts random pruning based on pre-calculated layer-wise sparsity which draws intuitive analogies from physics. (v) *Robust Bird* (RB), which can be regarded as an early stopped OMP. (vi) *Small Dense*. It is an important sanity check via considering smaller dense networks with the same parameter counts as the ones of sparse networks. Comprehensive results of these subnetworks at $80\%$ and $90\%$ sparsity are reported in Table 1, where the chosen sparsity follows routine options (Evci et al., 2020a; Liu et al., 2021b).

Table 1: Performance showing the appearance of poor robust generalization/robust overfitting, and the effectiveness of our sparse proposals with various comparisons to other sparsification methods on CFAIR-10 with ResNet-18. The difference between best and final robust accuracy indicates degradation in performance during training. We pick the best checkpoint by the best robust accuracy on the validation set. Bold numbers indicate superior performance, and $\downarrow$ displays shrunk robust generalization gap compared to dense models. Note that model picking criterion and the presentation style are consistent for all tables.

| Sparsity(%) | Settings | Robust Accuracy | | | Standard Accuracy | | | Training | Inference | Robust |
|---|---|---|---|---|---|---|---|---|---|---|
| | | Best | Final | Diff. | Best | Final | Diff. | FLOPs ($\times 10^{17}$) | FLOPs ($\times 10^{9}$) | Generalization |
| 0 | Baseline | 51.10 | 43.61 | 7.49 | 81.15 | 83.38 | −2.23 | 772.41 | 260.07 | 38.82 |
| 80 | Small Dense | 49.04 | 44.18 | 4.86 | 76.64 | 80.77 | −4.13 | 69.54 | 23.41 | 21.68 $\downarrow$ 17.14 |
| | Random Pruning | 49.32 | 43.97 | 5.35 | 77.75 | 81.27 | −3.52 | 154.40 | 51.99 | 25.70 $\downarrow$ 13.12 |
| | OMP | 50.16 | 45.02 | 5.14 | 79.80 | 82.39 | −2.59 | 966.63 | 65.39 | 28.38 $\downarrow$ 10.44 |
| | SNIP | 50.46 | 46.44 | 4.02 | 80.13 | 83.20 | −3.07 | 241.85 | 81.43 | 25.24 $\downarrow$ 13.58 |
| | GraSP | 50.16 | 45.31 | 4.85 | 78.38 | 82.42 | −4.04 | 187.11 | 63.00 | 26.28 $\downarrow$ 12.54 |
| | SynFlow | 51.17 | 46.91 | 4.26 | 79.08 | 83.19 | −4.11 | 256.09 | 86.23 | 24.66 $\downarrow$ 14.16 |
| | IGQ | 51.12 | 46.74 | 4.38 | 79.73 | 83.26 | −3.53 | 239.39 | 80.60 | 25.41 $\downarrow$ 13.41 |
| | Robust Bird | 50.18 | 46.10 | 4.08 | 78.46 | 82.42 | −3.96 | 209.54 | 64.64 | 23.37 $\downarrow$ 15.45 |
| | Flying Bird | 51.62 | 46.37 | 5.25 | 80.55 | 83.17 | −2.62 | 239.38 | 80.60 | 28.90 $\downarrow$ 9.92 |
| | Flying Bird+ | 51.70 | 47.51 | 4.19 | 80.74 | 83.16 | −2.42 | 120.04 | 40.42 | 23.89 $\downarrow$ 14.93 |
| 90 | Small Dense | 46.81 | 45.48 | 1.33 | 77.13 | 78.54 | −1.41 | 24.31 | 8.19 | 13.86 $\downarrow$ 24.96 |
| | Random Pruning | 47.09 | 44.97 | 2.12 | 75.25 | 78.77 | −3.52 | 77.16 | 25.98 | 15.11 $\downarrow$ 23.71 |
| | OMP | 49.31 | 46.11 | 3.20 | 77.99 | 81.00 | −3.01 | 877.76 | 35.47 | 19.05 $\downarrow$ 19.77 |
| | SNIP | 49.49 | 47.85 | 1.64 | 77.74 | 81.92 | −4.18 | 154.35 | 51.97 | 16.20 $\downarrow$ 22.62 |
| | GraSP | 48.56 | 46.80 | 1.76 | 79.02 | 81.39 | −2.37 | 113.38 | 38.18 | 16.80 $\downarrow$ 22.02 |
| | SynFlow | 50.08 | 48.02 | 2.06 | 81.15 | 81.56 | −0.41 | 156.74 | 52.77 | 14.68 $\downarrow$ 24.14 |
| | IGQ | 49.74 | 48.05 | 1.69 | 81.06 | 81.84 | −0.78 | 141.10 | 47.51 | 15.95 $\downarrow$ 22.87 |
| | Robust Bird | 49.09 | 46.56 | 2.53 | 77.96 | 80.93 | −2.97 | 133.42 | 39.01 | 16.62 $\downarrow$ 22.20 |
| | Flying Bird | 50.97 | 48.10 | 2.87 | 79.62 | 82.93 | −3.31 | 141.10 | 47.51 | 20.07 $\downarrow$ 18.75 |
| | Flying Bird+ | 50.88 | 49.27 | 1.61 | 79.95 | 82.65 | −2.70 | 66.67 | 22.45 | 15.16 $\downarrow$ 23.66 |

As shown in Table 1, we first observe the occurrence of poor robust generalization with 38.82% RA gap and robust overfitting with 7.49% RA degradation, when training the dense network (Baseline). Fortunately, coincided with our claims, injecting appropriate sparsity effectively tackle the issue. For instance, RB greatly shrinks the RGG by 15.45%/22.20% at 80/90% sparsity, while also mitigates robust overfitting by 2.53% ∼ 4.08%. Furthermore, comparing all *static* pruning methods, we find that (1) Small Dense and RP behave the worst, which suggests the identified sparse typologies play important roles rather than reduced network capacity only; (2) RB shows clear advantages to OMP in terms of all measurements, especially for 78.32% ∼ 84.80% training FLOPs savings. It validates our RB proposal that a few epochs of standard training are enough to learn a high-quality sparse structure for further robustification, and thus there is no need to complete the full training in the tickets finding stage like traditional OMP. (3) SynFlow and IGQ approaches have the best RA and SA, while RB obtains the superior robust generalization among static pruning approaches.

Finally, we explore the influence of training regimes during the RB ticket finding on CIFAR-100 with ResNet-18. Table A6 demonstrates that RB tickets perform best when found with the cheapest standard training. Specifically, at 90% and 95% sparsity, SGD RB tickets outperform both Fast AT (Wong et al., 2020) and PGD-10 RB tickets with up to 1.27% higher RA and 1.86% narrower RGG. Figure A7 offers a possible explanation for this phenomenon: the SGD training scheme more quickly develops high-level network connections, during the early epochs of training (Achille et al., 2019). As a result, RB Tickets pruned from the model trained with SGD achieve superior quality.

## 4.2 FLYING BIRD IS A BETTER BIRD

In this section, we discuss the advantages of dynamic sparsity and show that our Flying Bird(+) is a superior bird. Table 1 examines the effectiveness of FB(+) on CIFAR-10 with ResNet-18, and several consistent observations can be drawn: ❶ FB(+) achieve 9.92% ∼ 23.66% RGG reduction, 2.24% ∼ 5.88% decrease for robust overfitting, compared with the dense network. And FB+ at 80% sparsity even pushes the RA 0.60% higher. ❷ Although the smaller dense network shows the leading performance w.r.t improving robust generalization, the robustness has been largely sacrificed, with up to 4.29% RA degradation, suggesting that only reducing models' parameter counts is insufficient to keep satisfactory SA/RA. ❸ FB and FB+ achieve superior performance of RA for both the *best* and *final* checkpoints across all methods, including RB. ❹ Regardless of small dense and random pruning due to their poor robustness, FB+ reaches the most impressive robust generalization (rank #1 or #2) with the least training and inference costs. Precisely, FB+ obtains 84.46% ∼ 91.37% training FLOPs and 84.46% ∼ 93.36% inference FLOPs saving, i.e., *Flying Bird+ is SUPER light-weight*.

Table 2: Performance showing the effectiveness of our proposed approaches across different datasets with ResNet-18. The subnetworks at 80% sparsity are selected here.

| Dataset | Settings | Robust Accuracy | | | Standard Accuracy | | | Training | Inference | Robust |
|---|---|---|---|---|---|---|---|---|---|---|
| | | Best | Final | Diff. | Best | Final | Diff. | FLOPs ($\times 10^{17}$) | FLOPs ($\times 10^{9}$) | Generalization |
| CIFAR-10 | Baseline | 51.10 | 43.61 | 7.49 | 81.15 | 83.38 | −2.23 | 772.41 | 260.07 | 38.82 |
| | Robust Bird | 50.18 | 46.10 | 4.08 | 78.46 | 82.42 | −3.96 | 209.54 | 64.64 | 23.37 ↓15.45 |
| | Flying Bird | 51.62 | 46.37 | 5.25 | 80.55 | 83.17 | −2.62 | 239.38 | 80.60 | 28.90 ↓9.92 |
| | Flying Bird+ | 51.70 | 47.51 | 4.19 | 80.74 | 83.16 | −2.42 | 120.04 | 40.42 | 23.89 ↓14.93 |
| CIFAR-100 | Baseline | 26.93 | 19.62 | 7.31 | 52.03 | 53.91 | −1.88 | 772.41 | 260.07 | 54.56 |
| | Robust Bird | 25.54 | 20.82 | 4.72 | 48.79 | 53.33 | −4.54 | 189.80 | 58.00 | 25.46 ↓29.10 |
| | Flying Bird | 26.64 | 22.00 | 4.64 | 53.57 | 55.41 | −1.84 | 237.12 | 79.84 | 27.46 ↓27.10 |
| | Flying Bird+ | 26.66 | 23.37 | 3.29 | 52.29 | 55.23 | −2.94 | 100.90 | 33.97 | 20.12 ↓34.44 |
| Tiny-ImageNet | Baseline | 20.84 | 15.76 | 5.08 | 43.57 | 46.64 | −3.07 | 6179.30 | 1040.29 | 36.84 |
| | Robust Bird | 19.58 | 16.45 | 3.13 | 43.70 | 46.30 | −2.60 | 1410.44 | 215.15 | 15.22 ↓21.62 |
| | Flying Bird | 20.34 | 19.00 | 1.34 | 45.95 | 46.86 | −0.91 | 1884.01 | 317.17 | 14.93 ↓21.91 |
| | Flying Bird+ | 20.36 | 19.11 | 1.25 | 45.67 | 46.73 | −1.06 | 1225.80 | 206.36 | 13.24 ↓23.60 |

Table 3: Performance showing the effectiveness of our proposed approaches with other architectures, i.e., VGG-16 on CIFAR-10/100. The subnetworks at 80% sparsity are selected here.

| Architecture | Dataset | Settings | Robust Accuracy | | | Standard Accuracy | | | FLOPs | | Robust |
|---|---|---|---|---|---|---|---|---|---|---|---|
| | | | Best | Final | Diff. | Best | Final | Diff. | Training | Inference | Generalization |
| VGG-16 | CIFAR-10 | Baseline | 48.33 | 42.73 | 5.60 | 76.84 | 79.73 | −2.89 | 574.69 | 193.50 | 28.00 |
| | | Robust Bird | 47.69 | 41.66 | 6.03 | 75.32 | 78.58 | −3.26 | 165.95 | 51.48 | 23.57 ↓4.43 |
| | | Flying Bird | 48.43 | 44.65 | 3.78 | 77.53 | 79.72 | −2.19 | 173.56 | 58.44 | 21.01 ↓6.99 |
| | | Flying Bird+ | 48.25 | 45.24 | 3.01 | 77.48 | 79.55 | −2.07 | 94.63 | 31.86 | 17.75 ↓10.25 |
| VGG-16 | CIFAR-100 | Baseline | 22.76 | 18.06 | 4.70 | 46.11 | 46.88 | −0.77 | 574.69 | 193.50 | 63.18 |
| | | Robust Bird | 23.46 | 17.48 | 5.98 | 46.33 | 47.59 | −1.26 | 165.77 | 51.42 | 48.19 ↓14.99 |
| | | Flying Bird | 22.75 | 17.96 | 4.79 | 46.61 | 47.36 | −0.75 | 172.14 | 57.96 | 48.11 ↓15.07 |
| | | Flying Bird+ | 22.92 | 19.02 | 3.90 | 47.01 | 48.11 | −1.10 | 69.93 | 23.54 | 34.63 ↓28.55 |

**Superior performance across datasets and architectures.** We further evaluate the performance of FB(+) across various datasets (CIFAR-10, CIFAR-100 and Tiny-ImageNet) and architectures (ResNet-18 and VGG-16). Table 2 and 3 display that both static and dynamic sparsity of our proposals serve effective remedies for improving robust generalization and mitigating robust overfitting, with $4.43\% \sim 15.45\%$, $14.99\% \sim 34.44\%$ and $21.62\% \sim 23.60\%$ RGG reduction across different architectures on CIFAR-10, CIFAR-100 and Tiny-ImageNet, respectively. Moveover, both RB and FB(+) gain significant efficiency, with up to $87.83\%$ training and inference FLOPs savings.

**Superior performance across improved attacks.** Additionally, we verify both RB and FB(+) under improved attacks, i.e., Auto-Attack (Croce & Hein, 2020) and CW Attack (Carlini & Wagner, 2017). As shown in Table A8, our approaches shrink the robust generalization gap by up to $30.76\%$ on CIFAR-10/100, and largely mitigate robust overfitting. This piece of evidence shows our proposal's effectiveness sustained across diverse attacks.

**Combining FB+ with existing start-of-the-art (SOTA) mitigation.** Previous works (Chen et al., 2021e; Zhang et al., 2021a; Wu et al., 2020b) point out that smoothening regularizations (e.g., KD (Hinton et al., 2015) and SWA (Izmailov et al., 2018)) help robust generalization and lead to SOTA robust accuracies. We combine them with our FB+ and collect the robust accuracy on CIFAR-10 with ResNet-18 in Figure 4. The extra robustness gains from FB+ imply that they makes complementary contributions.

Figure 4: Combination of FB+ and previous SOTAs.

**Excluding obfuscated gradients.** A common "counterfeit" of robustness improvements is less effective adversarial examples resulted from obfuscated gradients (Athalye et al., 2018). Table A7 demonstrates the maintained enhanced robustness under unseen transfer attacks, which excludes the possibility of gradient masking. More are referred to Section A3.

### 4.3 ABLATION STUDY AND VISUALIZATION

**Different sparse initialization and update frequency.** As two major components in the dynamic sparsity exploration (Evci et al., 2020a), we conduct thorough ablation studies in Table 4 and 5. We found the performance of *Flying Bird+* is more sensitive to different sparse initialization; using SNIP to produce initial layer-wise sparsity and updating the connections per 2000 iterations serves the superior configuration for FB+.

Table 4: Ablation of different sparse initialization in Flying Bird+. Subnetwroks at 80% initial sparsity are chosen on CIFAR-10 with ResNet-18.

| Initialization | Robust Accuracy | | | Standard Accuracy | | | Robust Generalization |
|---|---|---|---|---|---|---|---|
| | Best | Final | Diff. | Best | Final | Diff. | |
| Uniform | 49.09 | 46.96 | 2.13 | 78.32 | 80.32 | −2.00 | 15.61 |
| ERK | 50.57 | 47.70 | 2.87 | 79.53 | 82.21 | −2.68 | 18.64 |
| SNIP | 51.30 | 49.17 | 2.13 | 79.86 | 82.28 | −2.42 | 15.15 |
| GraSP | 50.76 | 47.88 | 2.88 | 78.52 | 82.48 | −3.96 | 18.54 |
| SynFlow | 50.56 | 48.75 | 1.81 | 78.51 | 82.17 | −3.66 | 14.10 |
| IGQ | 50.88 | 49.27 | 1.61 | 79.95 | 82.65 | −2.70 | 15.16 |

Table 5: Ablation of different update frequency in Flying Bird+. Subnetworks at 80% initial sparsity are chosen on CIFAR-10 with ResNet-18.

| Update Frequency | Robust Accuracy | | | Standard Accuracy | | | Robust Generalization |
|---|---|---|---|---|---|---|---|
| (iterations) | Best | Final | Diff. | Best | Final | Diff. | |
| 100 | 50.32 | 49.02 | 1.30 | 81.28 | 81.99 | −0.71 | 13.36 |
| 500 | 50.57 | 48.37 | 2.20 | 79.76 | 82.73 | −2.97 | 18.92 |
| 1000 | 50.99 | 48.34 | 2.65 | 79.55 | 82.69 | −3.14 | 19.85 |
| 2000 | 51.19 | 48.39 | 2.80 | 79.80 | 83.00 | −3.20 | 19.17 |
| 5000 | 50.39 | 48.49 | 1.90 | 79.11 | 82.58 | −3.47 | 17.95 |
| 10000 | 50.08 | 48.02 | 2.06 | 79.25 | 82.50 | −3.25 | 17.64 |

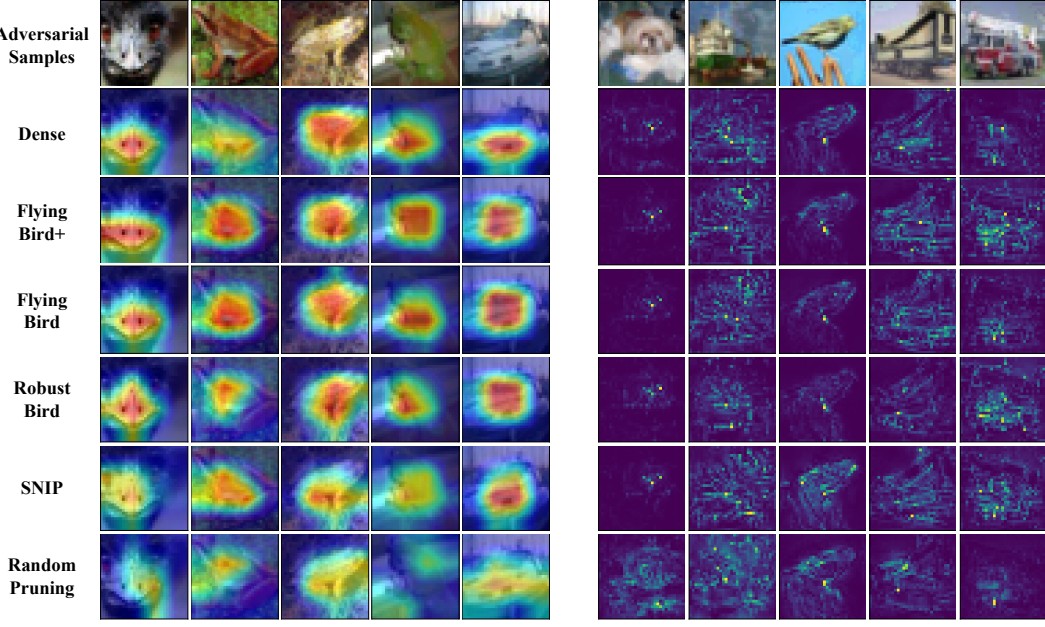

Figure 5: Loss landscape visualization of robusitified dense network and sparse networks (90% sparsity) from different sparsification approaches on CIFAR-10 with ResNet-18.

**Final checkpoint loss landscapes.** From visualizations in Figure 5, FB and FB+ converge to much flatter loss valleys, which evidences their effectiveness in closing robust generalization gaps.

**Attention and saliency maps.** To visually inspect the benefits of our proposal, here we provide attention and saliency maps generated by Grad-GAM (Selvaraju et al., 2017) and tools in (Smilkov et al., 2017). Comparing the dense model to our "talented birds" (e.g., FB+), Figure 6 shows that our approaches have enhanced concentration on main objects, and are capable of capturing more local feature information, aligning better with human perception.

Figure 6: (*Left*) Visualization of attention heatmaps on adversarial images based on Grad-Cam (Selvaraju et al., 2017). (*Right*) Saliency map visualization on adversarial samples (Smilkov et al., 2017).

## 5 CONCLUSION

We show the adversarial training of dense DNNs incurs a severe robust generalization gap, which can be effectively and efficiently resolved by injecting appropriate sparsity. Our proposed Robust Bird and Flying Bird(+) with static and dynamic sparsity, significantly mitigate the robust generalization gap while retaining competitive standard/robust accuracy, besides substantially reduced computation. Our future works plan to investigate channel- and block-wise sparse structures.

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

## A1 MORE TECHNIQUE DETAILS

**Algorithms of Robust Bird and Flying Bird(+).** Here we present the detailed procedure to identify robust bird and flying bird(+), as summarized in algorithm 1 and 2. Note that for the increasing frequency on Line 10 and 11 in algorithm 2, we compare the measurements stored in the queue between two consequent epochs and calculate the frequency of increasing.

---

**Algorithm 1:** Finding a Robust Bird

**Input:** $f(x; \theta_0)$ w. initialization $\theta_0$, target sparsity $s\%$, FIFO queue $Q$ with length $l$, threshold $\tau$
**Output:** Robust bird $f(x; m_{t^*} \odot \theta_T)$

1 **while** $t < t_{\max}$ **do**
2     Update network parameters $\theta_t \leftarrow \theta_{t-1}$ via *standard training*
3     Apply static pruning towards target sparsity $s\%$ and obtain the sparse mask $m_t$
4     Calculate the Hamming distance $\delta_H(m_t, m_{t-1})$, append result to $Q$
5     $t \leftarrow t + 1$
6     **if** $\max(Q) < \tau$ **then**
7        $t^* \leftarrow t$
8        Rewind $f(x; m_{t^*} \odot \theta_{t^*}) \rightarrow f(x; m_{t^*} \odot \theta_0)$
9        Training $f(x; m_{t^*} \odot \theta_0)$ via PGD-AT for T epochs
10        **return** $f(x; m_{t^*} \odot \theta_T)$
11     **end**
12 **end**

---

**Algorithm 2:** Finding a Flying Bird(+)

**Input:** Initialization parameters $\theta_0$, sparse masks $m$ of sparsity $s\%$, FIFO queue $Q_p$ and $Q_g$
       with length $l$, pruning and growth increasing ratio $\delta_p$ and $\delta_g$, update threshold $\epsilon$,
       optimize interval $\Delta t$, parameter update ratio $k\%$, ratio update starting point $t_{\text{start}}$
**Output:** Flying bird(+) $f(x; m \odot \theta_T)$

1 **while** $t < T$ **do**
2     Update network parameters $\theta_t \leftarrow \theta_{t-1}$ via PGD-AT;
3     # Record training statistics
4     Add robust generalization gap between train and validation set to $Q_p$
5     Add robust validation loss to $Q_g$
6     # Update sparse masks $m$
7     **if** *(t mod $\Delta t$)* == 0 **then**
8        |---Optional for Flying Bird+---|
9        # Update pruning and growth ratio $p\%$, $g\%$
10        **if** $t > t_{\text{start}}$ and increasing frequency of $Q_p \geq \epsilon$: $p = (1 + \delta_p) \times k$ **else** $p = k$
11        **if** $t > t_{\text{start}}$ and increasing frequency of $Q_g \geq \epsilon$: $g = (1 + \delta_g) \times k$ **else** $g = k$
12        |---Optional for Flying Bird+---|
13        *Prune $p\%$ parameters with smallest weight magnitude*
14        *Grow $g\%$ parameters with largest gradient*
15        Update sparse mask $m$ accordingly
16     **end**
17 **end**

---

## A2 MORE IMPLEMENTATION DETAILS

### A2.1 OTHER COMMON DETAILS

We select two checkpoints during training: *best*, which has the best RA values on the validation set, and *final*, i.e., the last checkpoint. And we report both RA and SA of these two checkpoints on test sets. Apart from the robust generalization gap, we also show the extent of robust overfitting numerically by the difference of RA between *best* and *final*. Furthermore, we calculate the FLOPs

at both training and inference stages to evaluate the prices of obtaining and exploiting the subnetworks respectively, in which we approximate the FLOPs of the back-propagation to be twice that of forwarding propagation (Yang et al., 2020).

### A2.2 MORE DETAILS ABOUT ROBUST BIRD

For the experiments of RB tickets finding, we comprehensively study three training regimes: standard training with stochastic gradient descent (SGD), adversarial training with PGD-10 AT (Madry et al., 2018b), and Fast AT (Wong et al., 2020). Following Pang et al. (2021), we train the network with an SGD optimizer of $0.9$ momentum and $5 \times 10^{-4}$ weight decay. We use a batch size of $128$. For the experiments of PGD-10 AT, we adopt the $\ell_\infty$ PGD attack with a maximum perturbation $\epsilon = 8/255$ and a step size $\alpha = 2/255$. And the learning rate starts from $0.1$, then decays by ten times at $50, 150$ epoch. As for fast AT, we use a cyclic schedule with a maximum learning rate equals $0.2$.

### A2.3 MORE DETAILS ABOUT FLYING BIRD(+)

For the experiments of Flying Bird+, the increasing ratio of pruning and growth $\delta_p, \delta_q$ is kept default to $0.4\%$ and $0.05\%$, respectively.

## A3 MORE EXPERIMENT RESULTS

### A3.1 MORE RESULTS ABOUT ROBUST BIRD

**Accuracy during RB Tickets Finding**    Figure A7 shows the curve of standard test accuracy during the training phase of RB ticket finding. We can observe the SGD training scheme develops high-level network connections much faster than the others, which provides a possible explanation for the superior quality of RB tickets from SGD.

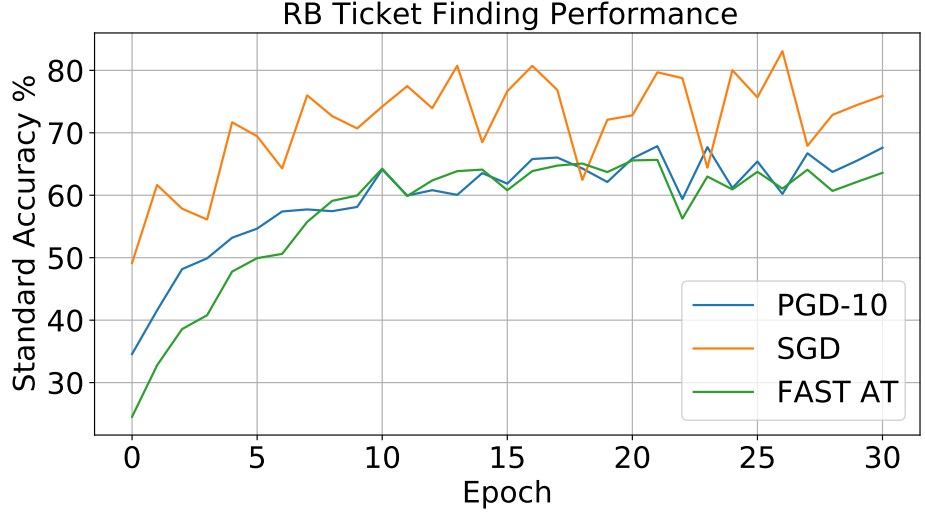

Figure A7: Standard accuracy (SA) of PGD-10, SGD, and Fast AT during the *RB ticket finding* phase.

**Mask Similarity Visualization.**    Figure A8 visualizes the dynamic similarity scores for each epoch among masks found via SGD, Fast AT, and PGD-10. Specifically, the similarity scores (You et al., 2020) reflect the Hamming distance between a pair of masks. We notice that masks found by SGD and PGD-10 share more common structures. A possible reason is that Fast AT usually adopts a cyclic learning rate schedule, while SGD and PGD use a multi-step decay schedule.

**Different training regimes for finding RB tickets.**    We denote the subnetworks identified by standard training with SGD, adversarial training with Fast AT (Wong et al., 2020) and adversarial train-

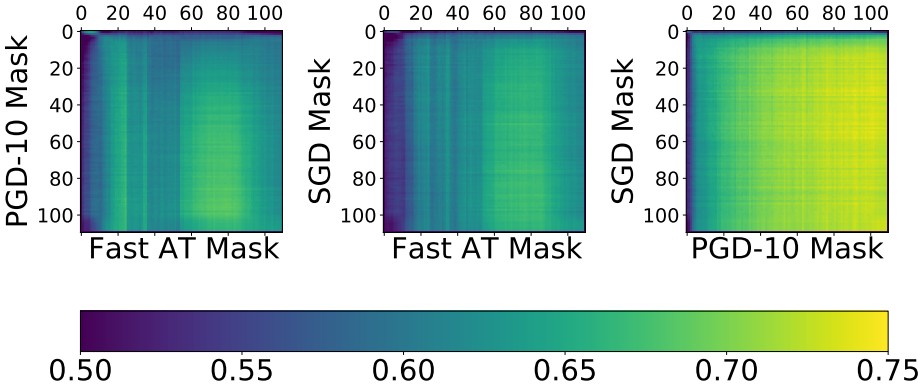

Figure A8: Similarity scores by epoch among masks found via Fast AT, SGD, and PGD-10. A brighter color denotes higher similarity.

Table A6: Comparison results of different training regimes for RB ticket finding on CIFAR-100 with ResNet-18. The subnetworks at $90\%$ and $95\%$ are selected here.

| Sparsity(%) | Settings | Roubst Accuarcy | | | Standard Accuarcy | | | Robust Generalization |
|---|---|---|---|---|---|---|---|---|
| | | Best | Final | Diff. | Best | Final | Diff. | |
| 0 | Baseline | 26.93 | 19.62 | 7.31 | 52.03 | 53.91 | −1.88 | 54.56 |
| 90 | SGD tickets | 25.83 | 23.40 | 2.43 | 49.35 | 53.51 | −4.16 | 18.37↓ 36.19 |
| | Fast AT tickets | 25.15 | 22.88 | 2.27 | 51.00 | 51.75 | −0.75 | 20.23↓ 34.33 |
| | PGD-10 tickets | 25.34 | 22.96 | 2.38 | 52.01 | 53.27 | −1.26 | 20.03↓ 34.53 |
| 95 | SGD tickets | 24.77 | 24.12 | 0.65 | 49.88 | 50.89 | −1.01 | 9.18↓ 45.38 |
| | Fast AT tickets | 23.50 | 22.46 | 1.04 | 41.67 | 43.19 | −1.52 | 9.53↓ 45.03 |
| | PGD-10 tickets | 24.44 | 23.77 | 0.67 | 49.30 | 50.65 | −1.35 | 9.86↓ 44.70 |

ing with PGD-10 AT as SGD tickets, Fast AT tickets, and PGD-10 tickets, respectively. Table A6 demonstrate the SGD tickets has the best performance.

**Loss Landscape Visualization**  We visualize the loss landscape of the dense network, random pruned subnetwork, and robust bird tickets at $30\%$ sparsity in Figure A9. Compared with the dense model and random pruned subnetwork, RB tickets found by the standard training shows much flatter loss landscapes, which provide a high-quality starting point for further robustification.

### A3.2   MORE RESULTS ABOUT FLYING BIRD(+)

**Excluding Obfuscated Gradients.**  To exclude this possibility of gradient masking, we show that our methods maintain improved robustness under unseen transfer attacks. As shown in Table A7, the left part represents the testing accuracy of perturbed test samples from an unseen robust model, and the right part shows the transfer testing performance on an unseen robust model (here we use a separately robustified ResNet-50 with PGD-10 on CIFAR-100).

**Performance under Improved Attacks.**  We report the performance of both RB and FB(+) under Auto-Attack (Croce & Hein, 2020) and CW Attack (Carlini & Wagner, 2017). For Auto-Attack, we keep the default setting with $\epsilon = \frac{8}{255}$. And for CW Attack we perform 1 search step on C with an initial constant of 0.1. And we use 100 iterations for each search step with the learning rate of 0.01. As shown in Table A8, both RB and FB(+) outperform the dense counterpart in terms of robust generalization. And FB+ achieves superior performance.

**More Datasets and Architectures**  We report more results of different sparsification methods across diverse datasets and architectures at Table A9, A10, A11 and  A12, from which we observe our approaches are capable of improving robust generalization and mitigating robust overfitting.

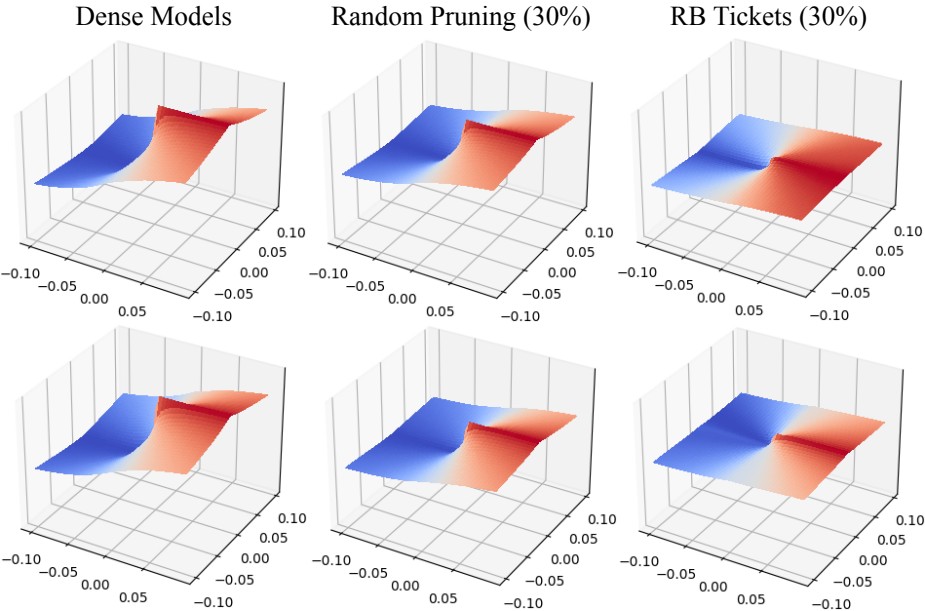

Figure A9: Loss landscapes visualizations (Engstrom et al., 2018; Chen et al., 2021e) of the dense model (unpruned), random pruned subnetwork at 30% sparsity, and Robust Bird (RB) tickets at 30% sparsity found by the *standard training*. The ResNet-18 backbone with *the same original initialization* on CIFAR-10 is adopted here. Results demonstrate that RB tickets offer a *smoother* and *flatter* starting point for further robustification in the second stage.

Table A7: Transfer attack performance from/on an unseen non-robust model, where the attacks are generated by/applied to the non-robust model. The robust generalization gap is also calculated based on transfer attack accuracies between train and test sets. We use ResNet-18 on CIFAR-10/100 and sub-networks at 80% sparsity.

| Dataset | Settings | Transfer Attack from Unseen Model | | | | Transfer Attack on Unseen Model | | | |
|---|---|---|---|---|---|---|---|---|---|
| | | Accuracy | | | Robust | Accuracy | | | Robust |
| | | Best | Final | Diff. | Generalization | Best | Final | Diff. | Generalization |
| CIFAR-10 | Baseline | 79.68 | 82.03 | −2.35 | 16.43 | 70.48 | 79.85 | −9.37 | 11.84 |
| | Robust Bird | 77.33 | 81.04 | −3.71 | 12.18 | 73.17 | 77.03 | −3.86 | 11.49 |
| | Flying Bird | 79.13 | 82.17 | −3.04 | 13.49 | 71.59 | 77.19 | −5.60 | 11.88 |
| | Flying Bird+ | 79.47 | 81.90 | −2.43 | 11.85 | 70.43 | 76.00 | −5.57 | 11.42 |
| CIFAR-100 | Baseline | 50.51 | 52.15 | −1.64 | 45.91 | 48.67 | 54.48 | −5.81 | 36.98 |
| | Robust Bird | 47.25 | 51.74 | −4.49 | 28.80 | 47.47 | 50.90 | −3.43 | 35.82 |
| | Flying Bird | 51.80 | 53.52 | −1.72 | 31.98 | 45.56 | 50.61 | −5.05 | 35.39 |
| | Flying Bird+ | 50.72 | 53.56 | −2.84 | 25.09 | 47.04 | 49.43 | −2.39 | 35.09 |

**Distributions of Adopted Sparse Initialization.** We report the layer-wise sparsity of different initial sparse masks. As shown in Figure A10, we observe that subnetworks generally have better performance when the top layers remain most of the parameters.

**Training Curve of Flying Bird+.** Figure A11 shows the training curve of Flying Bird+, in which the red dotted lines represent the time for increasing the pruning ratio and the green dotted lines for growth ratio. The detailed training curve demonstrates the flexibility of flying bird+ for dynamically adjusting the sparsity levels.

## A4 EXTRA RESULTS AND DISCUSSION

We sincerely appreciate all anonymous reviewers' and area chairs' constructive discussions for improving this paper. Extra results and discussions are presented in this section.

Table A8: Evaluation under improved attacks (i.e., Auto-Attack and CW-Attack) on CIFAR-10/100 with ResNet-18 at 80% sparsity. The robust generalization gap is computed under improved attacks.

| Dataset | Settings | Auto-Attack | | | | CW-Attack | | | |
|---|---|---|---|---|---|---|---|---|---|
| | | Accuracy | | | Robust | Accuracy | | | Robust |
| | | Best | Final | Diff. | Generalization | Best | Final | Diff. | Generalization |
| CIFAR-10 | Baseline | 47.41 | 41.59 | 5.82 | 35.30 | 75.76 | 66.13 | 9.63 | 30.39 |
| | Robust Bird | 45.90 | 42.45 | 3.45 | 21.58 ↓13.72 | 73.95 | 73.52 | 0.43 | 17.67 ↓12.72 |
| | Flying Bird | 47.55 | 43.57 | 3.98 | 26.55 ↓8.75 | 75.30 | 72.08 | 3.22 | 21.77 ↓8.62 |
| | Flying Bird+ | 47.06 | 44.09 | 3.17 | 21.73 ↓13.57 | 76.00 | 73.83 | 2.17 | 17.77 ↓12.62 |
| CIFAR-100 | Baseline | 23.16 | 17.68 | 5.48 | 49.73 | 45.83 | 36.21 | 9.62 | 57.52 |
| | Robust Bird | 21.29 | 18.00 | 3.29 | 21.72 ↓28.01 | 43.30 | 42.39 | 0.91 | 30.82 ↓26.70 |
| | Flying Bird | 22.74 | 19.44 | 3.30 | 25.18 ↓24.55 | 46.23 | 42.36 | 3.87 | 35.50 ↓22.02 |
| | Flying Bird+ | 22.90 | 20.31 | 2.59 | 19.05 ↓30.68 | 45.86 | 43.90 | 1.96 | 26.76 ↓30.76 |

Table A9: More results of different sparcification methods on CIFAR-10 with ResNet-18.

| Sparsity(%) | Settings | Robust Accuracy | | | Standard Accuracy | | | Robust |
|---|---|---|---|---|---|---|---|---|
| | | Best | Final | Diff. | Best | Final | Diff. | Generalization |
| 0 | Baseline | 51.10 | 43.61 | 7.49 | 81.15 | 83.38 | −2.23 | 38.82 |
| 95 | Small Dense | 45.99 | 44.55 | 1.44 | 74.26 | 75.64 | −1.38 | 7.87 ↓30.95 |
| | Random Pruning | 45.64 | 44.18 | 1.46 | 75.20 | 75.20 | 0.00 | 7.96 ↓30.86 |
| | OMP | 47.08 | 46.23 | 0.85 | 78.77 | 79.36 | −0.59 | 12.01 ↓26.81 |
| | SNIP | 48.18 | 46.72 | 1.46 | 78.55 | 79.21 | −0.66 | 9.58 ↓29.24 |
| | GraSP | 48.58 | 47.15 | 1.43 | 78.95 | 79.44 | −0.49 | 10.37 ↓28.45 |
| | SynFlow | 48.93 | 48.22 | 0.71 | 78.70 | 78.90 | −0.20 | 8.25 ↓30.57 |
| | IGQ | 48.82 | 47.56 | 1.26 | 79.44 | 79.76 | −0.32 | 9.33 ↓29.49 |
| | Robust Bird | 47.53 | 46.48 | 1.05 | 78.33 | 78.78 | −0.45 | 9.20 ↓29.62 |
| | Flying Bird | 49.62 | 48.46 | 1.16 | 78.12 | 81.43 | −3.31 | 13.32 ↓25.52 |
| | Flying Bird+ | 49.37 | 48.84 | 0.53 | 80.33 | 80.28 | 0.05 | 9.27 ↓29.55 |

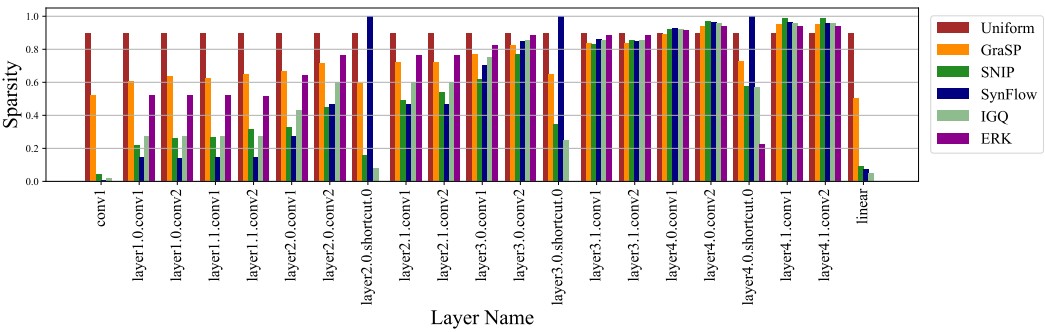

Figure A10: Layer-wise sparisty of different initial sparse masks with ResNet-18

### A4.1 MORE RESULTS OF DIFFERENT SPARSITY

We report more results of subnetworks with 40/60% sparsity on CIFAR-10/100 with ResNet-18 and VGG-16. As shown in Table A13, A14, A15 and A16, our flying bird(+) achieves consistent improvement than baseline unpruned networks, in terms of 2.45 ~ 19.81% narrower robust generalization gaps with comparable RA and SA performance.

### A4.2 MORE RESULTS ON WIDERESNET

We further evaluate our flying bird(+) with WideResNet-34-10 on CIFAR-10 and report the results on Table A17. We can observe that compared with the dense network, our methods significantly shrink the robust generalization gap by up to 13.14% and maintain comparable RA/SA performance.

Table A10: More results of different sparcification methods on CIFAR-10 with VGG-16.

| Sparsity(%) | Settings | Robust Accuracy | | | Standard Accuracy | | | Robust |
| | | Best | Final | Diff. | Best | Final | Diff. | Generalization |
|---|---|---|---|---|---|---|---|---|
| 0 | Baseline | 48.33 | 42.73 | 5.60 | 76.84 | 79.73 | −2.89 | 28.00 |
| 80 | Random Pruning | 46.14 | 40.33 | 5.81 | 74.42 | 76.68 | −2.26 | 21.01 ↓6.99 |
| | OMP | 47.90 | 43.19 | 4.71 | 76.60 | 80.02 | −3.42 | 24.97 ↓3.03 |
| | SNIP | 48.03 | 43.17 | 4.86 | 76.68 | 80.08 | −3.40 | 24.71 ↓3.29 |
| | GraSP | 47.91 | 42.34 | 5.57 | 75.74 | 78.87 | −3.13 | 23.65 ↓4.35 |
| | SynFlow | 48.47 | 45.32 | 3.15 | 77.62 | 79.09 | −1.47 | 20.17 ↓7.83 |
| | IGQ | 48.57 | 44.25 | 4.32 | 77.51 | 80.01 | −2.50 | 22.79 ↓5.21 |
| | Robust Bird | 47.69 | 41.66 | 6.03 | 75.32 | 78.58 | −3.26 | 23.57 ↓4.43 |
| | Flying Bird | 48.43 | 44.65 | 3.78 | 77.53 | 79.72 | −2.19 | 21.01 ↓6.99 |
| | Flying Bird+ | 48.25 | 45.24 | 3.01 | 77.48 | 79.55 | −2.07 | 17.75 ↓10.25 |
| 90 | Random Pruning | 44.33 | 40.33 | 4.00 | 71.27 | 74.46 | −3.19 | 15.48 ↓12.52 |
| | OMP | 47.84 | 43.34 | 4.50 | 75.60 | 79.10 | −3.50 | 18.29 ↓9.71 |
| | SNIP | 47.76 | 44.27 | 3.49 | 75.92 | 79.62 | −3.70 | 17.85 ↓10.15 |
| | GraSP | 45.96 | 42.12 | 3.84 | 75.19 | 77.03 | −1.84 | 15.04 ↓12.96 |
| | SynFlow | 47.54 | 45.79 | 1.75 | 78.43 | 78.70 | −0.27 | 14.40 ↓13.60 |
| | IGQ | 47.79 | 45.12 | 2.67 | 74.87 | 79.19 | −4.32 | 16.06 ↓11.94 |
| | Robust Bird | 47.09 | 44.13 | 2.96 | 75.53 | 78.36 | −2.83 | 16.57 ↓11.43 |
| | Flying Bird | 48.45 | 45.55 | 2.90 | 75.82 | 79.21 | −3.39 | 16.56 ↓11.44 |
| | Flying Bird+ | 48.39 | 46.26 | 2.13 | 78.73 | 79.12 | −0.39 | 12.47 ↓15.53 |

Table A11: More results of different sparcification methods on CIFAR-100 with ResNet-18.

| Sparsity(%) | Settings | Robust Accuracy | | | Standard Accuracy | | | Robust |
| | | Best | Final | Diff. | Best | Final | Diff. | Generalization |
|---|---|---|---|---|---|---|---|---|
| 0 | Baseline | 26.93 | 19.62 | 7.31 | 52.03 | 53.91 | −1.88 | 54.56 |
| 80 | Small Dense | 24.40 | 21.83 | 2.57 | 51.87 | 51.64 | 0.23 | 21.93 ↓32.63 |
| | Random Pruning | 25.92 | 20.83 | 5.09 | 48.16 | 51.31 | −3.15 | 34.04 ↓20.52 |
| | OMP | 25.12 | 20.18 | 4.94 | 50.08 | 52.81 | −2.73 | 28.57 ↓26.00 |
| | SNIP | 26.61 | 23.55 | 3.06 | 49.47 | 54.79 | −5.32 | 23.69 ↓30.87 |
| | GraSP | 25.37 | 20.79 | 4.58 | 50.27 | 53.29 | −3.02 | 28.03 ↓26.53 |
| | SynFlow | 26.31 | 23.52 | 2.79 | 48.33 | 54.49 | −6.16 | 20.29 ↓34.27 |
| | IGQ | 26.87 | 23.07 | 3.80 | 49.80 | 54.39 | −4.59 | 27.04 ↓27.52 |
| | Robust Bird | 25.54 | 20.82 | 4.72 | 48.79 | 53.33 | −4.54 | 25.46 ↓29.10 |
| | Flying Bird | 26.64 | 22.00 | 4.64 | 53.57 | 55.41 | −1.84 | 27.46 ↓27.10 |
| | Flying Bird+ | 26.66 | 23.37 | 3.29 | 52.29 | 55.23 | −2.94 | 20.12 ↓34.44 |
| 90 | Small Dense | 23.61 | 22.81 | 0.80 | 48.44 | 48.63 | −0.19 | 11.18 ↓43.38 |
| | Random Pruning | 24.06 | 21.45 | 2.61 | 47.06 | 49.73 | −2.67 | 18.04 ↓36.52 |
| | OMP | 24.45 | 21.38 | 3.07 | 48.02 | 51.26 | −3.24 | 17.11 ↓37.45 |
| | SNIP | 26.10 | 24.46 | 1.64 | 52.35 | 52.88 | −0.53 | 11.54 ↓43.02 |
| | GraSP | 24.83 | 22.74 | 2.09 | 51.09 | 52.55 | −1.46 | 14.55 ↓40.01 |
| | SynFlow | 25.45 | 24.62 | 0.83 | 51.03 | 51.96 | −0.93 | 10.38 ↓44.18 |
| | IGQ | 26.22 | 24.87 | 1.35 | 52.37 | 53.16 | −0.79 | 13.90 ↓40.66 |
| | Robust Bird | 24.65 | 22.96 | 1.69 | 46.16 | 51.87 | −5.71 | 16.14 ↓38.42 |
| | Flying Bird | 26.14 | 23.57 | 2.57 | 50.53 | 54.78 | −4.25 | 16.73 ↓37.83 |
| | Flying Bird+ | 26.26 | 24.16 | 2.10 | 51.16 | 53.97 | −2.81 | 11.44 ↓43.12 |

## A4.3    COMPARISON WITH EFFICIENT ADVERSARIAL TRAINING METHODS

To elaborate more about training efficiency, we compare our methods with two efficient training methods. Shafahi et al. (2019) proposed Free Adversarial Training that improves training efficiency by reusing the gradient information, which is orthogonal to our approaches and can be easily combined with our methods to pursue more efficiency by replacing the PGD-10 training with Free AT.

Table A12: More results of different sparcification methods on CIFAR-100 with VGG-16.

| Sparsity(%) | Settings | Robust Accuracy | | | Standard Accuracy | | | Robust Generalization |
|---|---|---|---|---|---|---|---|---|
| | | Best | Final | Diff. | Best | Final | Diff. | |
| 0 | Baseline | 22.76 | 18.06 | 4.70 | 46.11 | 46.88 | −0.77 | 63.18 |
| 80 | Random Pruning | 22.38 | 15.76 | 6.62 | 41.79 | 44.85 | −3.06 | 51.15 ↓12.03 |
| | OMP | 22.98 | 16.32 | 6.66 | 45.45 | 45.96 | −0.51 | 53.59 ↓9.59 |
| | SNIP | 23.34 | 17.83 | 5.51 | 46.58 | 48.55 | −1.97 | 40.42 ↓22.76 |
| | GraSP | 23.05 | 16.50 | 6.55 | 43.01 | 46.84 | −3.83 | 49.71 ↓13.47 |
| | SynFlow | 23.02 | 17.67 | 5.35 | 45.55 | 47.33 | −1.78 | 41.70 ↓21.48 |
| | IGQ | 23.60 | 17.44 | 6.16 | 45.77 | 47.43 | −1.66 | 48.18 ↓15.00 |
| | Robust Bird | 23.46 | 17.48 | 5.98 | 46.33 | 47.59 | −1.26 | 48.19 ↓15.00 |
| | Flying Bird | 22.75 | 17.96 | 4.79 | 46.61 | 47.36 | −0.75 | 48.11 ↓15.07 |
| | Flying Bird+ | 22.92 | 19.02 | 3.90 | 47.01 | 48.11 | −1.10 | 34.63 ↓28.55 |
| 90 | Random Pruning | 21.48 | 16.33 | 5.15 | 43.10 | 44.93 | −1.83 | 31.34 ↓31.84 |
| | OMP | 22.18 | 17.38 | 4.80 | 44.81 | 45.63 | −0.82 | 38.91 ↓24.27 |
| | SNIP | 22.92 | 20.30 | 2.62 | 48.50 | 49.05 | −0.55 | 20.02 ↓43.16 |
| | GraSP | 22.17 | 17.60 | 4.57 | 44.54 | 47.00 | −2.46 | 29.76 ↓33.42 |
| | SynFlow | 22.58 | 18.88 | 3.70 | 43.62 | 46.73 | −3.11 | 24.96 ↓38.22 |
| | IGQ | 22.55 | 18.56 | 3.99 | 44.96 | 48.08 | −3.12 | 27.91 ↓35.27 |
| | Robust Bird | 22.80 | 19.19 | 3.61 | 45.78 | 48.61 | −2.83 | 26.46 ↓36.72 |
| | Flying Bird | 23.59 | 18.86 | 4.73 | 46.64 | 48.45 | −1.81 | 34.05 ↓29.13 |
| | Flying Bird+ | 23.31 | 20.34 | 2.97 | 45.51 | 48.13 | −2.62 | 22.16 ↓41.02 |

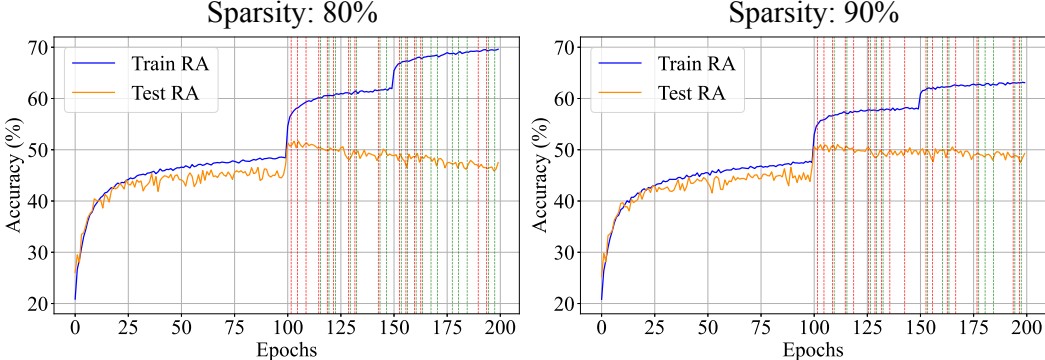

Figure A11: Training curve of Flying Bird+ at 80%(Left) and 90%(Right) sparsity on CIFAR-10 with ResNet-18. The Red and Green dotted lines indicate the time for increasing the pruning and growth ratio, respectively.

Table A13: Comparison results of the unpruned dense network and our flying birds at more sparsity levels. Experiments are conducted on CIFAR-10 with ResNet-18 under PGD-10 adversarial training.

| Sparsity% | Settings | Robust Accuracy | | | Standard Accuracy | | | Robust Generalization |
|---|---|---|---|---|---|---|---|---|
| | | Best | Final | Diff. | Best | Final | Diff. | |
| 0 | Baseline | 51.10 | 43.61 | 7.49 | 81.15 | 83.38 | −2.23 | 38.82 |
| 40 | Flying Bird+ | 51.25 | 43.45 | 7.80 | 81.51 | 82.94 | −1.43 | 34.38 ↓4.44 |
| 60 | Flying Bird | 51.20 | 43.58 | 7.62 | 81.27 | 83.35 | −2.08 | 35.65 ↓3.17 |
| | Flying Bird+ | 51.23 | 44.95 | 6.28 | 81.35 | 83.19 | −1.84 | 29.89 ↓8.93 |

Additionally, Li et al. (2020) uses magnitude pruning to locate sparse structures, which is similar to OMP reported in Table 1, except they use a smaller learning rate. Our methods achieve better performance and efficiency than OMP. Specifically, with 80% sparsity, our flying bird+ reaches a 4.49% narrower robust generalization gap and 1.54% higher RA yet only requires 87.58% less training FLOPs. Also, our methods can be easily combined with Fast AT for further training efficiency.

Table A14: Comparison results of the unpruned dense network and our flying birds at more sparsity levels. Experiments are conducted on CIFAR-100 with ResNet-18 under PGD-10 adversarial training.

| Sparsity% | Settings | Robust Accuracy | | | Standard Accuracy | | | Robust |
|---|---|---|---|---|---|---|---|---|
| | | Best | Final | Diff. | Best | Final | Diff. | Generalization |
| 0 | Baseline | 26.93 | 19.62 | 7.31 | 52.03 | 53.91 | −1.88 | 54.56 |
| 40 | Flying Bird | 26.63 | 19.80 | 6.83 | 53.44 | 54.46 | −1.02 | 48.66 ↓5.90 |
| | Flying Bird+ | 27.35 | 20.48 | 6.87 | 52.34 | 54.76 | −2.42 | 40.31 ↓14.25 |
| 60 | Flying Bird | 26.95 | 20.60 | 6.35 | 51.77 | 54.71 | −2.94 | 42.13 ↓12.43 |
| | Flying Bird+ | 26.95 | 21.38 | 5.57 | 51.77 | 55.32 | −3.55 | 34.75 ↓19.81 |

Table A15: Comparison results of the unpruned dense network and our flying birds at more sparsity levels. Experiments are conducted on CIFAR-10 with VGG-16 under PGD-10 adversarial training.

| Sparsity% | Settings | Robust Accuracy | | | Standard Accuracy | | | Robust |
|---|---|---|---|---|---|---|---|---|
| | | Best | Final | Diff. | Best | Final | Diff. | Generalization |
| 0 | Baseline | 48.33 | 42.73 | 5.60 | 76.84 | 79.73 | −2.89 | 28.00 |
| 40 | Flying Bird | 48.03 | 42.86 | 5.17 | 76.28 | 79.66 | −3.38 | 25.40 ↓2.60 |
| | Flying Bird+ | 49.13 | 43.56 | 5.57 | 77.03 | 79.92 | −2.89 | 23.19 ↓4.81 |
| 60 | Flying Bird | 48.06 | 43.69 | 4.37 | 78.31 | 80.11 | −1.80 | 25.55 ↓2.45 |
| | Flying Bird+ | 48.41 | 44.64 | 3.77 | 76.45 | 80.03 | −3.58 | 21.63 ↓6.37 |

Table A16: Comparison results of the unpruned dense network and our flying birds at more sparsity levels. Experiments are conducted on CIFAR-100 with VGG-16 under PGD-10 adversarial training.

| Sparsity% | Settings | Robust Accuracy | | | Standard Accuracy | | | Robust |
|---|---|---|---|---|---|---|---|---|
| | | Best | Final | Diff. | Best | Final | Diff. | Generalization |
| 0 | Baseline | 22.76 | 18.06 | 4.70 | 46.11 | 46.88 | −0.77 | 63.18 |
| 40 | Flying Bird | 23.22 | 18.20 | 5.02 | 45.20 | 46.95 | −1.75 | 59.19 ↓3.99 |
| | Flying Bird+ | 23.21 | 17.90 | 5.31 | 45.20 | 47.13 | −1.93 | 49.40 ↓13.78 |
| 60 | Flying Bird | 23.53 | 18.14 | 5.99 | 46.03 | 46.90 | −0.87 | 51.77 ↓11.41 |
| | Flying Bird+ | 23.61 | 17.91 | 5.70 | 46.17 | 47.59 | −1.42 | 49.78 ↓13.40 |

Table A17: Comparison results of the unpruned dense network and our flying birds on CIFAR-10 with WideResNet-34-10.

| Sparsity% | Settings | Robust Accuracy | | | Standard Accuracy | | | Robust |
|---|---|---|---|---|---|---|---|---|
| | | Best | Final | Diff. | Best | Final | Diff. | Generalization |
| 0 | Baseline | 54.73 | 46.83 | 7.90 | 84.08 | 85.84 | −1.76 | 52.60 |
| 80 | Flying Bird | 55.34 | 46.79 | 8.55 | 83.76 | 85.93 | −2.17 | 49.41 ↓3.19 |
| | Flying Bird+ | 55.34 | 46.82 | 8.52 | 83.76 | 85.97 | −2.21 | 46.73 ↓5.87 |
| 90 | Flying Bird | 54.27 | 46.16 | 8.11 | 85.44 | 86.01 | −0.57 | 45.41 ↓7.19 |
| | Flying Bird+ | 54.24 | 46.91 | 7.33 | 85.52 | 85.93 | −0.41 | 39.46 ↓13.14 |

### A4.4 COMPARISON WITH OTHER PRUNING AND SPARSE TRAINING METHODS

Compared with the recent work (Özdenizci & Legenstein, 2021), our flying bird(+) is different at both levels of goal and methodologies. Firstly, Özdenizci & Legenstein (2021) pursues a superior adversarial robust testing accuracy for sparsely connected networks. While we aim to investigate the relationship between sparsity and robust generalization, and demonstrate that introducing appropriate sparsity (e.g., LTH-based static sparsity or dynamic sparsity) into adversarial training

substantially alleviates the robust generalization gap and maintains comparable or even better standard/robust accuracies. Secondly, Özdenizci & Legenstein (2021) samples network connectivity from a learned posterior to form a sparse subnetwork. However, our flying bird first removes the parameters with the lowest magnitude, which ensures a small term of the first-order Taylor approximation of the loss and thus limits the impact on the output of networks (Evci et al., 2020a). And then, it allows new connectivity with the largest gradient to grow to reduce the loss quickly (Evci et al., 2020a). Furthermore, we propose an enhanced variant of Flying Bird, i.e., Flying Bird+, which not only learns the sparse topologies but also is capable of adaptively adjusting the network capacity to determine the right parameterization level "on-demand" during training, while Özdenizci & Legenstein (2021) stick to a fixed parameter budget.

Another work, HYDRA (Sehwag et al., 2020) also has several differences from our robust birds. Specifically, HYDRA starts from a robust pre-trained dense network, which requires at least hundreds of epochs for adversarial training. However, our robust bird's pre-training only needs a few epochs of standard training. Therefore, Sehwag et al. (2020) has significantly higher computational costs, compared to ours. Then, Sehwag et al. (2020) adopt TRADES (Zhang et al., 2019) for adversarial training, which also requires auxiliary inputs of clean images, while our methods follow the classical adversarial training (Madry et al., 2018b) and only take adversarial perturbed samples as input. Moreover, for CIFAR-10 experiments, Sehwag et al. (2020) uses 500k additional pseudo-labeled images from the Tiny-ImageNet dataset with a robust semi-supervised training approach. However, all our methods and experiments do not leverage any external data.

Furthermore, one concurrent work (Fu et al., 2021) demonstrates that there exist subnetworks with inborn robustness. Such randomly initialized networks have matching or even superior robust accuracy of adversarially trained networks with similar parameter counts. It's interesting to utilize this finding for further improvement of robust generalization, and we will investigate it in future works.

