# OpenReview forum: "Sparsity Winning Twice: Better Robust Generalization from More Efficient Training"
_ICLR.cc/2022/Conference — ICLR 2022 Poster_

### Official Review · Reviewer_dw7x · 2021-11-01

**Correctness:** 4
**Technical Novelty And Significance:** 3
**Empirical Novelty And Significance:** 4
**Recommendation:** 6
**Confidence:** 4

**Main Review:**

(+) The idea is very interesting and promising. It is reasonable to find sparsity helps both reduce overfitting and improve training efficiency. For adversarial training, that could lead to very significant cost reduction.

(+) The authors’ study on how to inject sparsity is very comprehensive. They considered two alternatives for sparse adversarial training: a static Robust Bird (RB) training, and a dynamic Flying Bird (FB) training. The former identifies critical mask structure at early training stage, while the latter continues to optimize the mask throughout the entire training.

(+) It is great the authors show that their proposed methods can be combined to boost previous SOTAs. That definitely amplifies their work’s value.

(+/-) Experiments are a bit limited, using only two models (VGG and Res18). However, the aspects being evaluated as well as the ablations are very thorough, and I kinda agree the results mostly suffice to validate their points.

(-) However, the biggest issue I have with experiments is: they did not compare with latest efficient adversarial training methods, such as “Adversarial training for free!”, NeurIPS 2019; and “Towards practical lottery ticket hypothesis for adversarial training” in arXiv 2020. The latter one is also based on sparsity and has certain overlap with Robust Bird. The authors did compare with Fast AT, but that was placed in the Appendix only.

(-) Furthermore, even standard adversarial training or adversarial pruning methods could have been made cheaper, by either early-stopping with less epochs, or using fast/free-AT to replace their more expensive AT sub-modules. The authors need to discuss whether those methods can become their competitive baselines, in a convincing way, not ignoring it.

(-) It is unclear to me whether the final robustness gain comes from the good mask structure or just sparsity itself. In particular, random pruning in Table 1 performs better than I would expect, inviting the aforementioned question.

(-) More sparsity levels should have been tried.


**Summary Of The Paper:**

This paper studies an important topic of improving adversarial training with different sparsity forms, and the authors made positive discoveries that injecting sparsity properly would make a win-win between efficiency and generalization.

All findings are not too surprising, but the authors’ effort of presenting a particularly comprehensive empirical study is acknowledged.


**Summary Of The Review:**

My biggest reservation currently lies in lacking comparison with SOTA efficient adversarial training methods. The value of this work cannot be solely established without that comparison. Overall, I’m currently rating as borderline and would decide my final rating based on the authors’ rebuttal.

---

> ### Author Response · Authors · 2021-11-19
> **Response for Reviewer dw7x [Cons1-4]**
>
> Thanks for rating our work as interesting and comprehensive. To address your concerns, we provide point-wise responses as below.
>
> **[Cons1: More architecture]** To further evaluate our approaches, we conduct more experiments on WideResNet-34-10 and report the results in Table S1. We can observe that compared with the unpruned network, our flying birds(+) significantly shrink the robust generalization gap by 3.19%-13.14% and maintain comparable robust and standard accuracy at the same time. We have included the new results in Section A4 of our updated version, which is highlighted in red color.
>
> Table S1 Comparison results of the unpruned dense network and our flying birds on CIFAR-10 with WideResNet-34-10.
>
> | Sparsity |Settings| RA Best | RA Final | RA Diff. | SA Best | SA Final | SA Diff. | Robust Generalization |
> | :------: | :------ |:-----: | :------: | :------: | :-----: | :------: | :------: | :-------------------: |
> |    0     | Baseline | 54.73  |  46.83   |   7.90    |  84.08  |  85.84   |  -1.76   |  52.60   |
> |    80    | Flying Bird |  55.34    |    46.79    |    8.55    |   83.76    |    85.93    |    -2.17    |      49.41 (↓3.19)     |
> |    80    | Flying Bird+ |  55.34    |    46.82    |    8.52    |   83.76    |    85.97    |    -2.21    |      46.73 (↓5.87)     |
> |    90    | Flying Bird | 54.27    |    46.16    |    8.11    |   85.44    |    86.01    |    -0.57    |      45.41 (↓7.19)     |
> |    90    | Flying Bird+ |  54.24    |    46.91    |    7.33    |   85.52    |    85.93    |    -0.41    |     39.46 (↓13.14)     |
>
> **[Cons2: Compare with efficient adversarial training methods]** We respectfully disagree. Actually, the main focus of our work is mitigating the robust generalization gap via injecting appropriate forms of sparsity. And efficiency is a byproduct due to sparsity itself.
> Moreover, the algorithm proposed in [Shafahi 2019] is orthogonal to our approaches and can be easily combined with our methods to pursue more efficiency by replacing the PGD-10 training with Free AT. [Li 2020] also uses magnitude pruning to locate sparse structures, which is similar to OMP reported in Table1, except they use a smaller learning rate. Our methods achieve better performance and efficiency than OMP. Specifically, with 80% sparsity, our flying bird+ reaches a 4.49% narrower robust generalization gap and 1.54% higher RA yet only requires 87.58% less training FLOPs.
> As pointed out by review dw7x, we also show our approaches can be easily combined with Fast AT, and we cite all aforementioned efficient adversarial training methods with a detailed discussion about the difference from our algorithms.
>
> **[Cons3: Combine with early-stopping and fast/free AT]** Firstly, we clarify that the topmost target of our work is improving robust generalization. And we have already adopted early-stopping in our experiments, where we pick the checkpoint with the best robust accuracy on the validation set (mentioned in Section A2.1 and Table1). Also, fast/free-AT are orthogonal to our approaches and can easily be plugged in for training both baseline unpruned networks and our sparse networks (robust birds and flying birds(+)). And the training cost (FLOPs) of baseline dense networks and our methods will also decrease proportionally. Thus they do not affect our conclusions. We cite all aforementioned publications with a detailed discussion in our revision.
>
>
> **[Cons4: Sparsity or Good sparse structure?]** Thanks for the question. Firstly, sparsity itself can effectively regularize the learning of over-parameterized neural networks and then benefit robust generalization[Balda 2019].
> However, as shown in Table 1, only sparsity is not enough. The first evidence is that when we remove the sparse structure inside the subnetwork and only maintain the parameter counts (denote as small dense), the network only reaches 46.81-49.04% RA, which suggests that the sparse structure matters.
> Secondly, we keep the sparse structure and compare masks identified from different methods. We can observe that the performance of sparse structures from random pruning is actually much poorer than our Flying Bird + with 2.38-3.79% lower RA and up to 1.81% larger generalization gaps. Thus the quality of sparse structures is also important for robustness and robust generalization.
> Additionally, from OMP, SNIP, SynFlow to our Flying Bird+, the increasing trend of RA from 50.16~51.70% further demonstrates that a high-quality sparse pattern benefits robustness.
>
> [Shafahi 2019] Adversarial Training for Free.\
> [Li 2020] Towards Practical Lottery Ticket Hypothesis for Adversarial Training.\
> [Balda 2019] Adversarial risk bounds for neural networks through sparsity based compression.

---

> ### Author Response · Authors · 2021-11-19
> **Response for Reviewer dw7x [Cons5]**
>
> **[Cons5: More sparsity levels]** Thanks for the suggestion, and we have conducted further experiments at more sparsity levels, i.e., 40% and 60% on CIFAR-10/100 with ResNet-18 and VGG-16. We report all results in Table S2-5, from which we observe a consistent improvement of our methods. Specifically, our flying birds achieve a 2.45%~19.81% narrower robust generalization gap and maintain comparable RA/SA performance. During the period of rebuttal, we have tried our best to report more results of two extra sparsity (40/60%), different architectures/datasets, and nearly comprehensive methods. We have included all new results and discussions in Section A4 of our revision, which is highlighted in red color.
>
> Table S2 Comparison results of the unpruned dense network and our flying birds at more sparsity levels. Experiments are conducted on CIFAR-10 with ResNet-18 under PGD-10 adversarial training.
>
> | Sparsity |Settings| RA Best | RA Final | RA Diff. | SA Best | SA Final | SA Diff. | Robust Generalization |
> | :------: | :------ |:-----: | :------: | :------: | :-----: | :------: | :------: | :-------------------: |
> |    0     | Baseline | 51.10  |  43.61   |   7.49    |  81.15  |  83.38   |  -2.23   |      38.82   |
> |    40    | Flying Bird+ |  51.25  |  43.45   |   7.80    |  81.51  |  82.94   |  -1.43   |     34.38 (↓4.44)     |
> |    60    | Flying Bird |  51.20   |  43.58   |   7.62   |  81.27  |  83.35   |  -2.08   |     35.65 (↓3.17)     |
> |    60    | Flying Bird+ |  51.23  |  44.95   |   6.28   |  81.35  |  83.19   |  -1.84   |     29.89 (↓8.93)     |
>
> Table S3 Comparison results of the unpruned dense network and our flying birds at more sparsity levels. Experiments are conducted on CIFAR-100 with ResNet-18 under PGD-10 adversarial training.
>
> | Sparsity |Settings| RA Best | RA Final | RA Diff. | SA Best | SA Final | SA Diff. | Robust Generalization |
> | :------: | :------ |:-----: | :------: | :------: | :-----: | :------: | :------: | :-------------------: |
> |    0     | Baseline | 26.93  |  19.62   |   7.31    |  52.03  |  53.91   |  -1.88   |      54.56   |
> |    40    | Flying Bird |  26.63    |    19.80    |    6.83    |   53.44    |    54.46    |    -1.02    |          48.66 (↓5.90)     |
> |    40    | Flying Bird+ |  27.35    |    20.48    |    6.87    |   52.34    |    54.76    |    -2.42    |          40.31 (↓14.25)     |
> |    60    | Flying Bird |  26.95    |    20.60    |    6.35    |   51.77    |    54.71    |   -2.94     |          42.13 (↓12.43)     |
> |    60    | Flying Bird+ |  26.95    |    21.38    |    5.57    |   51.77    |    55.32    |    -3.55    |          34.75 (↓19.81)     |
>
> Table S4 Comparison results of the unpruned dense network and our flying birds at more sparsity levels. Experiments are conducted on CIFAR-10 with VGG-16 under PGD-10 adversarial training.
>
> | Sparsity |Settings| RA Best | RA Final | RA Diff. | SA Best | SA Final | SA Diff. | Robust Generalization |
> | :------: | :------ |:-----: | :------: | :------: | :-----: | :------: | :------: | :-------------------: |
> |    0     | Baseline | 48.33  |  42.73   |   5.60    |  76.84  |  79.73   |  -2.89   |      28.00   |
> |    40    | Flying Bird |  48.03    |    42.86    |    5.17    |   76.28    |    79.66    |    -3.38    |         25.40 (↓2.60)     |
> |    40    | Flying Bird+ |  49.13  |  43.56   |   5.57   |  77.03  |  79.92   |  -2.89   |     23.19 (↓4.81)     |
> |    60    | Flying Bird | 48.06    |    43.69    |    4.37    |   78.31    |    80.11    |    -1.80    |          25.55 (↓2.45)     |
> |    60    | Flying Bird+ |  48.41    |    44.64    |    3.77    |   76.45    |    80.03    |    -3.58    |          21.63 (↓6.37)     |
>
> Table S5 Comparison results of the unpruned dense network and our flying birds at more sparsity levels. Experiments are conducted on CIFAR-100 with VGG-16 under PGD-10 adversarial training.
>
> | Sparsity |Settings| RA Best | RA Final | RA Diff. | SA Best | SA Final | SA Diff. | Robust Generalization |
> | :------: | :------ |:-----: | :------: | :------: | :-----: | :------: | :------: | :-------------------: |
> |    0     | Baseline | 22.76  |  18.06   |   4.70    |  46.11  |  46.88   |  -0.77   |      63.18   |
> |    40    | Flying Bird |  23.22    |    18.20    |    5.02    |   45.20    |    46.95    |    -1.75    |          59.19 (↓3.99)     |
> |    40    | Flying Bird+ |  23.21    |    17.90    |    5.31    |   45.20    |    47.13    |    -1.93    |          49.40 (↓13.78)     |
> |    60    | Flying Bird | 23.53    |    18.14    |    5.99    |   46.03    |    46.90    |    -0.87    |          51.77 (↓11.41)     |
> |    60    | Flying Bird+ |  23.61    |    17.91    |    5.70    |   46.17    |    47.59    |    -1.42    |          49.78 (↓13.40)     |

---

### Official Review · Reviewer_75p4 · 2021-11-01

**Correctness:** 4
**Technical Novelty And Significance:** 3
**Empirical Novelty And Significance:** 4
**Recommendation:** 8
**Confidence:** 4

**Main Review:**

Strength:
-	The motivation for introducing sparsity to adversarial training for both generalization and efficiency gains are valuable and novel. The paper investigates two sparsity forms: static and dynamic.
-	For the static form, the paper extends the early-bird idea by You et. al. 2020, and show the phenomenon also exists in the adversarial training scheme. So in general the novelty of this paper is only fair, except a surprise finding that even in adversarial training, EB tickets can still be drawn from a cheap standard pre-training stage.
-	For the dynamic form, the authors presented a vanilla version plus an advanced variant capable of adaptively adjusting the sparsity levels.
-	Experiments are solid and convincing. Besides PGD, the authors also used Auto-Attack (Croce & Hein, 2020) and CW Attack (Carlini & Wagner, 2017) for a more rigorous evaluation. The authors also carefully excluded obfuscated gradients using transferred unseen attacks
-	Attention visualization is another interesting and novel angle to compare different pruning methods.
-	Paper is structured well and easy to follow. I especially like that rationale is always clearly presented in company with the algorithms
-	All codes are included, and the reproducibility looks good to me.

Weakness:
-	I would appreciate if the authors could elaborate more on why the sparse mask found from a non-robust model could be reused to training a robust model. If that is true, I wonder whether or not there indeed exists any tight coupling between sparse mask structure and robustness.
-	Table 1 only have two sparsity levels: 80% and 90%. Why only this two, are they specifically cherry-picked? It would be better if the authors could demonstrate some more sparsity levels.
-	Table 1 should also have compared with Early Bird (You et al. (2020) ) and existing dynamic sparse training methods (Evci et al. (2020a); Liu et al. (2021b))
-	This paper does not provide any theoretical analysis on why the proposed strategy should work for efficient adversarial training. It is unclear which factor is the main performance booster: sparsity regularizing the adversarial overfitting, or sparsity for efficient “lottery”-style training. Despite this work being mainly empirical, some theory probing would have improved it.
-	Typos: “much more flatter” -> much flatter, and more. Proofreading is required.


**Summary Of The Paper:**

The authors proposed to leverage static and dynamic sparsity in efficient robust training. The proposed methods can significantly mitigate the robust generalization gap while retaining competitive performance (standard/robust accuracy) with substantially reduced computation budgets.

**Summary Of The Review:**

This is an interesting work with solid execution. More experimental clarification and analysis would be welcome.

---

> ### Author Response · Authors · 2021-11-19
> **Response for Reviewer 75p4 [Cons1]**
>
> Many thanks to Reviewer 75p4 for acknowledging our work as valuable and convincing. To address your concerns, we provide point-wise answers and extra experiment results as below. \
> **[Cons1. More discussion on sparse structure and robustness]**
> Firstly, [Li 2020, Wang 2020] seem to suggest that the sparse mask found from a robust model could be reused to train a robust subnetwork. To elaborate why the sparse mask from the standard training (SGD) regime is also the case, we calculate the similarity between sparse masks identified from different training schemes. Figure A9 shows the dynamic similarity scores (w.r.t. hamming distance) for each epoch among masks found via SGD, Fast AT, and PGD-10. We can observe that the masks from SGD share high similarities with masks from PGD-10, which provides more insights for the good performance of SGD masks.
>
> Secondly, we think the sparse topologies do have connections with robust generalization. As shown in Figure A10, high-quality sparse structures provide flatter starting points that benefit the following robust training. Also, as evidenced by our results in Table 1, with the same sparsity level, sparse structures located from diverse methods perform differently under the same training settings. Specifically, the subnetwork with 80% sparsity from random pruning only achieves 49.32% Robust Accuracy(RA) and 25.70% robust generalization gap, while with a higher-quality sparse structure, our robust birds reach 50.18% RA with a narrower robust generalization gap of 23.37%.
>
>  [Li 2020] Towards Practical Lottery Ticket Hypothesis for Adversarial Training.\
> [Wang 2020] Achieving Adversarial Robustness via Sparsity.

---

> > ### Comment · Reviewer_75p4 · 2021-11-30
> > **Post Rebuttal Update**
> >
> > I have read the rebuttal and other reviewers' comments. Overall the rebuttal has properly addressed my initial concerns and I will increase my score to 8.

---

> > > ### Author Response · Authors · 2021-11-30
> > > **Many thanks for increasing our score!**
> > >
> > > Dear Reviewer **75p4**,
> > >
> > > We really appreciate reviewer **75p4** for increasing our score and supporting the acceptance of our paper. We are glad to see our response has addressed reviewer **75p4**'s concerns.
> > >
> > > We are again very thankful for your time and all constructive feedback!
> > >
> > > Best wishes,
> > >
> > > Authors

---

> ### Author Response · Authors · 2021-11-19
> **Response for Reviewer 75p4 [Cons2]**
>
> **[Cons2. More sparsity levels]**
> We respectfully disagree. Our reported sparsity levels are not from cherry-picking. Actually, 80/90% sparsity levels are the common settings in the sparse training literature, e.g., Figure 2 in [Evci 2020] and Figure 3 in [Liu 2021]. And we follow these settings rather than cherry-picking. Moreover, to further evaluate our approaches, we conduct more experiments with other sparsity levels (40/60%) on CIFAR-10/100 with ResNet-18 and VGG-16. As shown in Table S1-4, our approaches yield consistent improvement compared with the dense counterpart, in terms of 2.45~19.81% narrower robust generalization gap with comparable RA/SA performance. We have tried our best to report more results in the period of rebuttal. In the aforementioned four tables, we provide the results of two extra sparsity (40/60%) and nearly comprehensive approaches in four settings of various architectures and datasets. All new results and discussions have been included in Section A4 of our revision, which is highlighted in red color.
>
> Table S1 Comparison results of the unpruned dense network and our flying birds at more sparsity levels. Experiments are conducted on CIFAR-10 with ResNet-18 under PGD-10 adversarial training.
>
> | Sparsity |Settings| RA Best | RA Final | RA Diff. | SA Best | SA Final | SA Diff. | Robust Generalization |
> | :------: | :------ |:-----: | :------: | :------: | :-----: | :------: | :------: | :-------------------: |
> |    0     | Baseline | 51.10  |  43.61   |   7.49    |  81.15  |  83.38   |  -2.23   |      38.82   |
> |    40    | Flying Bird+ |  51.25  |  43.45   |   7.80    |  81.51  |  82.94   |  -1.43   |     34.38 (↓4.44)     |
> |    60    | Flying Bird |  51.20   |  43.58   |   7.62   |  81.27  |  83.35   |  -2.08   |     35.65 (↓3.17)     |
> |    60    | Flying Bird+ |  51.23  |  44.95   |   6.28   |  81.35  |  83.19   |  -1.84   |     29.89 (↓8.93)     |
>
> Table S2 Comparison results of the unpruned dense network and our flying birds at more sparsity levels. Experiments are conducted on CIFAR-100 with ResNet-18 under PGD-10 adversarial training.
>
> | Sparsity |Settings| RA Best | RA Final | RA Diff. | SA Best | SA Final | SA Diff. | Robust Generalization |
> | :------: | :------ |:-----: | :------: | :------: | :-----: | :------: | :------: | :-------------------: |
> |    0     | Baseline | 26.93  |  19.62   |   7.31    |  52.03  |  53.91   |  -1.88   |      54.56   |
> |    40    | Flying Bird |  26.63    |    19.80    |    6.83    |   53.44    |    54.46    |    -1.02    |          48.66 (↓5.90)     |
> |    40    | Flying Bird+ |  27.35    |    20.48    |    6.87    |   52.34    |    54.76    |    -2.42    |          40.31 (↓14.25)     |
> |    60    | Flying Bird |  26.95    |    20.60    |    6.35    |   51.77    |    54.71    |   -2.94     |          42.13 (↓12.43)     |
> |    60    | Flying Bird+ |  26.95    |    21.38    |    5.57    |   51.77    |    55.32    |    -3.55    |          34.75 (↓19.81)     |
>
> Table S3 Comparison results of the unpruned dense network and our flying birds at more sparsity levels. Experiments are conducted on CIFAR-10 with VGG-16 under PGD-10 adversarial training.
>
> | Sparsity |Settings| RA Best | RA Final | RA Diff. | SA Best | SA Final | SA Diff. | Robust Generalization |
> | :------: | :------ |:-----: | :------: | :------: | :-----: | :------: | :------: | :-------------------: |
> |    0     | Baseline | 48.33  |  42.73   |   5.60    |  76.84  |  79.73   |  -2.89   |      28.00   |
> |    40    | Flying Bird |  48.03    |    42.86    |    5.17    |   76.28    |    79.66    |    -3.38    |         25.40 (↓2.60)     |
> |    40    | Flying Bird+ |  49.13  |  43.56   |   5.57   |  77.03  |  79.92   |  -2.89   |     23.19 (↓4.81)     |
> |    60    | Flying Bird | 48.06    |    43.69    |    4.37    |   78.31    |    80.11    |    -1.80    |          25.55 (↓2.45)     |
> |    60    | Flying Bird+ |  48.41    |    44.64    |    3.77    |   76.45    |    80.03    |    -3.58    |          21.63 (↓6.37)     |
>
> [Evci 2020] Rigging the lottery: Making All Tickets Winners.\
> [Liu 2021] Do we actually need dense over-parameterization? In-Time Over-Parameterization in Sparse Training.

---

> ### Author Response · Authors · 2021-11-19
> **Response for Reviewer 75p4 [Cons2] (Continued)**
>
> Table S4 Comparison results of the unpruned dense network and our flying birds at more sparsity levels. Experiments are conducted on CIFAR-100 with VGG-16 under PGD-10 adversarial training.
>
> | Sparsity |Settings| RA Best | RA Final | RA Diff. | SA Best | SA Final | SA Diff. | Robust Generalization |
> | :------: | :------ |:-----: | :------: | :------: | :-----: | :------: | :------: | :-------------------: |
> |    0     | Baseline | 22.76  |  18.06   |   4.70    |  46.11  |  46.88   |  -0.77   |      63.18   |
> |    40    | Flying Bird |  23.22    |    18.20    |    5.02    |   45.20    |    46.95    |    -1.75    |          59.19 (↓3.99)     |
> |    40    | Flying Bird+ |  23.21    |    17.90    |    5.31    |   45.20    |    47.13    |    -1.93    |          49.40 (↓13.78)     |
> |    60    | Flying Bird | 23.53    |    18.14    |    5.99    |   46.03    |    46.90    |    -0.87    |          51.77 (↓11.41)     |
> |    60    | Flying Bird+ |  23.61    |    17.91    |    5.70    |   46.17    |    47.59    |    -1.42    |          49.78 (↓13.40)     |

---

> ### Author Response · Authors · 2021-11-19
> **Response for Reviewer 75p4 [Cons3-5]**
>
> **[Cons3. More comparison with other methods]** Thanks for the suggestion, and we conduct new experiments of Early Bird Tickets [You 2019] and RigL [Evci 2020, Liu 2021] under PGD-10 adversarial training on CIFAR-10 with ResNet-18. With 80% sparsity, Early Bird Tickets and RigL achieve 48.62% and 50.41% RA, respectively, which are 3.08% and 1.29% lower than our flying bird+.
>
> **[Cons4. Lack of theoretical analysis]** For the rationale of robust bird, [Zhang 2021] shows theoretical justification that sparse winning tickets expand the convex region near the good local minima and provide a flatter loss landscape. Also, the flatness of the loss landscape is often believed to indicate the standard generalization. [Wu 2020, Andriushchenko 2017] suggest that a flatter adversarial loss landscape can also improve the robust generalization. They provide intuitions towards why robust birds can obtain well-generalizable robust models.
>
> For the rationale of flying bird (+), [Evci 2020] (Figure 6 left) shows that allowing new connections to grow offers more flexibility in navigating the loss landscape. Specifically, the authors investigate the loss landscape between two solutions, which are found by static sparse training and pruning, respectively. Whether using linear interpolation or finding a quadratic/cubic B ́ezier curve between the two solutions, no path without a high-loss barrier can be found. These observations suggest that the static sparse training algorithms may get stuck at local minima. Hence we propose flying bird, which optimizes the sparse structures simultaneously and creates the opportunity to escape bad local minima and search for the optimal sparse connectivity.
>
> Additionally, sparsity itself helps reduce adversarial overfitting. Take the result of 80% sparsity on CIFAR-10 with ResNet-18 as an example. As shown in Table1, the subnetwork from random pruning mitigates adversarial overfitting by 2.14% less RA degradation than the baseline unpruned network. Moreover, appropriate sparsity brings extra benefits. Specifically, LTH sparsity and dynamic sparsity training alleviate adversarial overfitting by up to 1.27% less RA degradation than random pruning.
>
> **[Cons5. Typos]** Thanks for pointing it out, and we have addressed all typos in the updated draft.
>
>
> [Evci 2020] Rigging the lottery: Making All Tickets Winners.\
> [Liu 2021] Do we actually need dense over-parameterization? In-Time Over-Parameterization in Sparse Training.\
> [You 2019] Drawing early-bird tickets: Towards more efficient training of deep networks.\
> [Zhang 2021] Why Lottery Ticket Wins? A Theoretical Perspective of Sample Complexity on Sparse Neural Networks.

---

### Official Review · Reviewer_a6jd · 2021-11-02

**Correctness:** 4
**Technical Novelty And Significance:** 3
**Empirical Novelty And Significance:** 4
**Recommendation:** 8
**Confidence:** 4

**Main Review:**

It is known that good sparsity can help prevent overfitting as well as reduce inference costs. The main barrier of making sparsity practical for (adversarial) training is the good sparsity pattern itself can be expensive to retrieve. This paper lays out a series of options to mitigate that barrier for adversarial training (AT).

The authors first demonstrate that good sparse subnetworks can be identified at the very early AT training stage with one-shot pruning, and the remaining stage could focus on training that very compact subnetwork. While similar observations were already drawn by You et. al. 2020 in the standard training, a notably progress made by the authors is that the mask can be located from just the cheap standard training, and it will still incur almost no performance loss when being re-used towards adversarial re-training.  I find it quite intriguing and meaningful.

The authors continue to investigate the role of sparsity when the sparse connections of subnetworks are on the fly. It allows more flexibility for the network to lower the training loss more, by tweaking not only weights but also topology. The authors further relaxed it to dynamically adjust the network capacity/sparsity to pursue superior robust performance, at a minor sacrifice of training efficiency.

The authors reported a variety of experiments using two backbones and three datasets. Their proposed methods are found to reduce robust generalization gap and overfitting by 34.44% and 4.02%, with comparable robust/standard accuracy boosts and 87.83%/87.82% training/inference FLOPs savings on CIFAR-100 with ResNet18; and similar competitive performance on other settings. Their proposed approaches can be combined with existing regularizers to yield new state-of-the-art results. The authors also carefully examined their robustness gains against adaptive and transferred attacks.



Here I have only two nitpicks. First, it would be better if the authors can give some theoretical insights why their strategy works, even with simplified assumptions. Currently the rationales offered are a bit vague. Second, while the FLOPS reduction is impressive, no actual hardware measurements were reported like in the Early Bird paper. Discussing the practical hardware benefits of the proposed strategy would enhance this work’s impact.

=================== post rebuttal ====================

I have read the rebuttal and the rebuttal addresses my concerns.

**Summary Of The Paper:**

Recent studies demonstrate adversarial training suffers from severe overfitting besides getting very expensive. This paper proposes to handle the two problems organically altogether, with the tool of sparse training.

The authors show that injecting appropriate sparsity forms in training could substantially shrink the robust generalization gap and alleviate the robust overfitting, meanwhile significantly saving training and inference FLOPs.


**Summary Of The Review:**

I am inclined to recommend acceptance, owing to the paper’s novel angle and solid experiment evaluations. I have no major critique.

---

> ### Author Response · Authors · 2021-11-19
> **Response for Reviewer a6jd**
>
> Thanks for rating our work as novel and solid. And we provide point-wise responses to your insightful comments as below.
>
> **[Cons1. Lack of theoretical justification]** For the rationale of robust bird, [Zhang 2021] provides theoretical evidence that sparse winning tickets expand the convex region near the good local minima, which results in a flatter loss landscape. While the flatness of the loss landscape is often believed to indicate the standard generalization, [Wu 2020, Andriushchenko 2017] suggest that a flatter adversarial loss landscape also improves the robust generalization. They lend intuitions towards why robust birds can obtain well-generalizable robust models.
>
> For the rationale of flying bird (+), [Evci 2020] (Figure 6 left) demonstrates that allowing new connections to grow provides more flexibility in navigating the loss landscape. Specifically, the authors examine the loss landscape between a solution found by static sparse training and a solution found by pruning. Whether using linear interpolation or finding a quadratic/cubic B ́ezier curve between the two solutions, no path without a high-loss barrier can be found. These observations suggest that the static sparse training algorithms can get stuck at local minima. Thus we propose flying bird, which optimizes the sparse structures simultaneously and creates the opportunity to escape bad local minima and search for the optimal sparse connectivity.
>
> **[Cons2. Lack of hardware measurements]** Since all our experiments are implemented based on unstructured pruning (a common practice of dynamic sparse training nowadays), the practical benefits on hardware platforms can be demonstrated, with the help of specialized packages. For example, in the range of 70%-90% unstructured sparsity, XNNPACK [Elsen 2020] has already shown 1.5 to 2 times speedups over dense baselines on smartphone processors. The hardware implementation of our algorithms is beyond the current work’s scope, but it would be interesting future work for us.
>
>
> [Zhang 2021] Why Lottery Ticket Wins? A Theoretical Perspective of Sample Complexity on Sparse Neural Networks.\
> [Wu 2020] Revisiting loss landscape for adversarial robustness.\
> [Andriushchenko 2017] Formal guarantees on the robustness of a classifier against adversarial manipulation.\
> [Evci 2020] Rigging the lottery: Making All Tickets Winners.\
> [Elsen 2020] Fast sparse convnets.

---

### Official Review · Reviewer_EaMs · 2021-11-05

**Correctness:** 3
**Technical Novelty And Significance:** 2
**Empirical Novelty And Significance:** 2
**Recommendation:** 5
**Confidence:** 3

**Main Review:**


It is well known that sparsity helps generalizing: from example, already Han 2015 in Fig 5 shows that pruning (simple L2 regularization + thresholding) helps the network generalizing over unseen data. Similarly, the same article shows that refining the surviving parameters after pruning yields better performance, and according to my personal experience it is also true that allowing parameters to enter and exit the pruning pool, i e allowing the pruning mask to evolve, improves performance under multiple points. So, in general those "findings" claimed by the article are not that novel according to the existing literature.
Concerning the experiments, the authors compare with a number of different reference strategies for pruning ratios of 80% and 90%. While I do not put into question the tradeoff between sparsity and performance, the performance corresponding to the sparsity ratio selected by the authors is so low and far beyond typical useful accuracy numbers for unpruned architecture that I cannot avoid questioning the meaningfulness of the reported results. In other words, it is inconclusive for the reader to see  that the proposed method performs x% better than the closest reference when you are in the 40/50% accuracy range for cifar 10.  I would suggest the authors to compare for a sparsity so that the  performance is  closer to more typical values for unpruned architectures.
In this context, it is not clear how a reference strategy based on simple L2 regularization without pruning would perform in term of generalization ability and that would be an interesting extra reference to consider.



[Han 2015] Song Han, Jeff Pool, John Tran, and William Dally. Learning both weights and connections for efficient neural network. In Advances in neural information processing systems, pp. 1135–1143,
2015b

**Summary Of The Paper:**

This paper deals with the problem of training a neural network so that it generalizes well over data unseen at training time. Namely, they address the particular case where a network is trained over an adversarial scheme. This paper proposes two methods for learning a sparse architecture called robust and flying bird. These methods aim at identifying sparse subnetworks arising during early training stages, so to get a pruning mask that eventually yields a sparse architecture (RobustBird). FlyingBird improves over Robust Bird in teh sense that the learning mask can be dynamically adjusted over time, i.e. pruned params may be recovered later on. The authors then experiment at training multiple architectures over different datasets in the experimental section, showing better generalization abilities and lower computational complexity (MACs) for their proposed methods Robust and Flying Birds. the authors conclude that sparsity help networks to generalize better, and as a byproduct it slashes computational complexity.


**Summary Of The Review:**

This paper addresses a relevant issue, however the main conclusions they draw didn't exactly make me jump on my seat, and while the proposed method may have some originality, the experimental results leave quite a question open about their significance given the considered sparsity/performance tradeoff point the authors consider. I would recommend the authors to revisit their claims under the light of the existing literature dealing with the relationship between sparsity and generalization ability, and to present their experimental results at a higher accuracy target.

---

> ### Author Response · Authors · 2021-11-19
> **Response for Reviewer EaMs [Cons1]**
>
> **[Cons1. Lack of novelty]** We respectfully argue our conclusions are novel and fundamentally different from existing literature.
>
> Our setting and problems are novel. First, [Han 2015] focuses on improving test accuracy and efficiency tradeoff on the clean test dataset. And for the standard training of a highly overparameterized model, there is almost no overfitting, as demonstrated in [Zhang 2016, Neyshabur 2017, Belkin 2019]. On the contrary, we investigate a totally different setup of adversarial training which suffers from robust overfitting and severe robust generalization gaps defined as the gap of robust accuracy (RA) between adversarial training and testing sets (See Evaluation Metrics in Section 4). Our goal is much more challenging: we aim to close the robust generalization gap with appropriate sparse topologies, and meantime, find a superior tradeoff among robust accuracy, standard accuracy (SA), and efficiency. And to our best knowledge, no existing work has introduced sparsity to alleviate robust generalization gaps. Furthermore, as shown in [Rice 2020], common methods for mitigating standard overfitting, e.g., explicit regularizations ($\ell_1$,$\ell_2$) and data augmentation, are much less effective in adversarial training, which also indicate the difference between our paper and previous literature.
>
> Moreover, our flying brid investigates sparse training (during-training pruning and growing) rather than post-training pruning in [Han 2015]. As evidenced in our Section 4, the approach used to inject appropriate forms of sparsity is highly important. And it remains a mystery whether we can utilize dynamic sparsity to improve robust generalization, which is also different from existing works.
>
> [Han 2015] Learning both weights and connections for efficient neural networks. \
> [Zhang 2016] Understanding deep learning requires rethinking generalization.\
> [Neyshabur 2017] Exploring generalization in deep learning.\
> [Belkin 2019] Reconciling modern machine learning practice and the classical bias-variance trade-off.\
> [Rice 2020] Overfitting in adversarially robust deep learning.

---

> ### Author Response · Authors · 2021-11-19
> **Response for Reviewer EaMs [Cons2]**
>
> **[Cons2. Low performance and inappropriate sparsity levels]**
> We first respectfully argue that our results are comparable with unpruned architecture rather than “low and far beyond … unpruned architecture.” As evidenced in Table1, 2, and 3, the baselines are the unpruned architecture, which performs on a par with our flying bird+ with 80% sparsity in terms of both standard accuracy (SA) and adversarial robust accuracy (RA). Specifically, in Table 3, the (SA, RA) of baseline unpruned model v.s. the (SA, RA) of our flying bird+ with 80% sparsity = (81.15, 51.10) v.s. (80.74, 51.70), where our sparse models have better robust accuracy. It supports that our reported results and tradeoff between sparsity and performance are meaningful and impressive. In addition, the sparsity levels (i.e., 80%/90%) we selected are the conventional settings in the sparse training literature, e.g., Figure 2 in [Evci 2020] and Figure 3 in [Liu 2021].
>
> Secondly, we want to further clarify the evaluation metrics of standard accuracy (SA) and adversarial robust accuracy (RA). Under the adversarial training context, SA is evaluated on a clean testing dataset and achieves around 80% on CIFAR-10 with ResNet-18; RA is evaluated on a challenge adversarial testing dataset with worst-case perturbations, and current state-of-the-art models (unpruned) can only achieve 50+% accuracy on CIFAR-10 with ResNet-18 under PGD-20 attacks. For example, with the same dataset and network backbone, [Chen 2021] (Table1) reported 80.78% SA and 50.72% RA under PGD-20; also, [Pang 2021] (Table3) reported 82.52% SA and 53.58% RA under PGD-10 (fewer steps of PGD attack leads to a slightly higher accuracy). In other words, our reported ~50% robust accuracy (RA) and ~80% standard accuracy (SA) actually reaches the current state-of-the-art results, as evidenced by the above literature.
>
> Thirdly, we also provide extra sparsity/performance tradeoffs, as you suggested. Particularly, as shown in Table S1-4, compared with the unpruned network, our flying birds obtain consistent improvement in terms of 2.45-19.81% narrower robust generalization gap and maintain comparable RA/SA in the meantime. During the time of the rebuttal period, we have tried our best to provide more results. In our four tables, we report the results of two extra sparsity (40/60%) and nearly comprehensive methods, with different architectures and datasets. We have included the new results and discussion in Section A4 of our updated manuscript, which is highlighted in red color.
>
>
> Table S1 Comparison results of the unpruned dense network and our flying birds at more sparsity levels. Experiments are conducted on CIFAR-10 with ResNet-18 under PGD-10 adversarial training.
>
> | Sparsity |Settings| RA Best | RA Final | RA Diff. | SA Best | SA Final | SA Diff. | Robust Generalization |
> | :------: | :------ |:-----: | :------: | :------: | :-----: | :------: | :------: | :-------------------: |
> |    0     | Baseline | 51.10  |  43.61   |   7.49    |  81.15  |  83.38   |  -2.23   |      38.82   |
> |    40    | Flying Bird+ |  51.25  |  43.45   |   7.80    |  81.51  |  82.94   |  -1.43   |     34.38 (↓4.44)     |
> |    60    | Flying Bird |  51.20   |  43.58   |   7.62   |  81.27  |  83.35   |  -2.08   |     35.65 (↓3.17)     |
> |    60    | Flying Bird+ |  51.23  |  44.95   |   6.28   |  81.35  |  83.19   |  -1.84   |     29.89 (↓8.93)     |
>
> Table S2 Comparison results of the unpruned dense network and our flying birds at more sparsity levels. Experiments are conducted on CIFAR-100 with ResNet-18 under PGD-10 adversarial training.
>
> | Sparsity |Settings| RA Best | RA Final | RA Diff. | SA Best | SA Final | SA Diff. | Robust Generalization |
> | :------: | :------ |:-----: | :------: | :------: | :-----: | :------: | :------: | :-------------------: |
> |    0     | Baseline | 26.93  |  19.62   |   7.31    |  52.03  |  53.91   |  -1.88   |      54.56   |
> |    40    | Flying Bird |  26.63    |    19.80    |    6.83    |   53.44    |    54.46    |    -1.02    |          48.66 (↓5.90)     |
> |    40    | Flying Bird+ |  27.35    |    20.48    |    6.87    |   52.34    |    54.76    |    -2.42    |          40.31 (↓14.25)     |
> |    60    | Flying Bird |  26.95    |    20.60    |    6.35    |   51.77    |    54.71    |   -2.94     |          42.13 (↓12.43)     |
> |    60    | Flying Bird+ |  26.95    |    21.38    |    5.57    |   51.77    |    55.32    |    -3.55    |          34.75 (↓19.81)     |
>
> [Evci 2020] Rigging the lottery: Making All Tickets Winners.\
> [Liu 2021] Do we actually need dense over-parameterization? In-Time Over-Parameterization in Sparse Training.\
> [Chen 2021] Robust Overfitting may be mitigated by properly learned smoothening.\
> [Pang 2021] Bag of Tricks for Adversarial Training.

---

> ### Author Response · Authors · 2021-11-19
> **Response for Reviewer EaMs [Cons2] (Continued)**
>
> Table S3 Comparison results of the unpruned dense network and our flying birds at more sparsity levels. Experiments are conducted on CIFAR-10 with VGG-16 under PGD-10 adversarial training.
>
> | Sparsity |Settings| RA Best | RA Final | RA Diff. | SA Best | SA Final | SA Diff. | Robust Generalization |
> | :------: | :------ |:-----: | :------: | :------: | :-----: | :------: | :------: | :-------------------: |
> |    0     | Baseline | 48.33  |  42.73   |   5.60    |  76.84  |  79.73   |  -2.89   |      28.00   |
> |    40    | Flying Bird |  48.03    |    42.86    |    5.17    |   76.28    |    79.66    |    -3.38    |         25.40 (↓2.60)     |
> |    40    | Flying Bird+ |  49.13  |  43.56   |   5.57   |  77.03  |  79.92   |  -2.89   |     23.19 (↓4.81)     |
> |    60    | Flying Bird | 48.06    |    43.69    |    4.37    |   78.31    |    80.11    |    -1.80    |          25.55 (↓2.45)     |
> |    60    | Flying Bird+ |  48.41    |    44.64    |    3.77    |   76.45    |    80.03    |    -3.58    |          21.63 (↓6.37)     |
>
>
> Table S4 Comparison results of the unpruned dense network and our flying birds at more sparsity levels. Experiments are conducted on CIFAR-100 with VGG-16 under PGD-10 adversarial training.
>
> | Sparsity |Settings| RA Best | RA Final | RA Diff. | SA Best | SA Final | SA Diff. | Robust Generalization |
> | :------: | :------ |:-----: | :------: | :------: | :-----: | :------: | :------: | :-------------------: |
> |    0     | Baseline | 22.76  |  18.06   |   4.70    |  46.11  |  46.88   |  -0.77   |      63.18   |
> |    40    | Flying Bird |  23.22    |    18.20    |    5.02    |   45.20    |    46.95    |    -1.75    |          59.19 (↓3.99)     |
> |    40    | Flying Bird+ |  23.21    |    17.90    |    5.31    |   45.20    |    47.13    |    -1.93    |          49.40 (↓13.78)     |
> |    60    | Flying Bird | 23.53    |    18.14    |    5.99    |   46.03    |    46.90    |    -0.87    |          51.77 (↓11.41)     |
> |    60    | Flying Bird+ |  23.61    |    17.91    |    5.70    |   46.17    |    47.59    |    -1.42    |          49.78 (↓13.40)     |

---

> ### Author Response · Authors · 2021-11-19
> **Response for Reviewer EaMs [Cons3]**
>
> **[Cons3. More Comparison with $\ell_2$ regularization]** Thanks for the suggestion. Actually, [Rice 2020] (Table2) has already studied the effect of $\ell_2$ regularization for mitigating robust generalization gap and showed $\ell_2$ regularization did not provide significant benefits, which is quite different from standard training. In our adversarial training experiments, we have already used $\ell_2$ regularization, which is also known as weight decay (we use 5e-4 of weight decay in the SGD optimizer). For further comparison, we conduct a new experiment on CIFAR-10 with the unpruned ResNet-18, without $\ell_2$ regularization. Results show that removing $\ell_2$ regularization causes 2.34% RA degradation and a 6.59% larger robust generalization gap. Overall, $\ell_2$ regularization brings marginal benefits in closing the robust generalization gap, which is already adopted in both baseline and our proposed methods.
>
> [Rice 2020] Overfitting in adversarially robust deep learning.

---

> ### Author Response · Authors · 2021-11-23
> **Sincerely expecting further discussions from Reviewer EaMs**
>
> Dear Reviewer EaMs,
>
> We thank reviewer EaMs time for the review and constructive comments. We really hope to have a further discussion with reviewer EaMs to see if our response solves the concerns.
>
> We would sincerely appreciate it if reviewer EaMs could reply to the most important points in our rebuttal. For example, as we pointed out, under **adversarial training**, ~50% robust accuracy (RA) and ~80% standard accuracy (SA) actually reach the current state-of-the-art results for ResNet-18 on CIFAR-10, as evidenced by the literature. Compared to unpruned dense architectures, our sparse proposals actually achieve comparable or even better RA and SA, which indicates a meaningful and impressive tradeoff between performance and efficiency.
>
> We genuinely hope reviewer EaMs could kindly check our response. Thank you!
>
> Best wishes,
>
> Authors

---

> ### Author Response · Authors · 2021-11-25
> **Sincerely expecting further discussions from Reviewer EaMs**
>
> Dear Reviewer **EaMs**,
>
> We sincerely hope to have further discussion with reviewer **EaMs** to see if our response solves his/her concerns. We are confident that our response should have cleared the air, and we can clarify more if there is more need. We are happy to answer any additional questions and provide more information.
>
> We genuinely hope reviewer EaMs could kindly check our response. Thank you!
>
> Best wishes,
>
> Authors

---

> ### Author Response · Authors · 2021-11-28
> **Sincerely expecting further discussions from Reviewer EaMs**
>
> Dear Reviewer **EaMs**,
>
> We really appreciate your time and constructive reviews.
>
> We politely send you a kind reminder that the discussion period is ending within 48 hours. Detailed replies and new experiments are provided to specifically address your concerns. A chance for further discussion with you is precious to us to see if our responses solve your concerns.
>
> Given all the new experiments and replies, are you willing to reconsider your rating? Your support is very important to us and we greatly appreciate that!
>
> Best Regards,
>
> Authors

---

> ### Author Response · Authors · 2021-11-29
> **Sincerely expecting further discussions from Reviewer EaMs**
>
> Dear Reviewer **EaMs**,
>
> We really appreciated your time and constructive reviews. We politely send you a kind reminder that the discussion period is ending within **24** hrs.
>
> Currently, the other three reviewers (**a6jd**, **75p4**, and **dw7x**) had acknowledged the contributions of this work, and they had made a consensus of the acceptance. Given our detailed replies and new experiments, could you please kindly check our response to see if it solves your concerns, so that you could also support our acceptance? Your support is very important to us and we greatly appreciate that!
>
> Meantime, please do not hesitate to reach out to us if there are other clarifications or experiments we can provide. Many thanks!
>
> Best Regards,
>
> Authors

---

### Public Comment · ~Ozan_Ozdenizci1 · 2021-11-15
**Comments**

Dear authors,

Thank you for the interesting work. In case it was overlooked by the authors and reviewers, we wanted to point to several pieces of relevant and recently published studies that were not mentioned.

Our recent work [1] published at ICML2021 similarly introduced a dynamic adversarial training framework to enable end-to-end sparse and robust training while keeping a static sparsity ratio. Authors' work, however, proposes a similar sparse adversarial training idea to differently focus on a robust generalization gap metric, which can be also evaluated for comparisons with [1]. Interestingly benign/robust accuracy comparisons for FlyingBird seem to converge to similar ranges at the first glance (e.g., sparsity ratios at 90% for VGG16 on CIFAR-10, as well as ResNet18 on CIFAR-100 are comparable to the networks in [2], however rigorous evaluations with, e.g., AutoAttack, would be necessary).

We also believe it would be relevant that the authors compare the results found via RobustBird (which has a parallel goal as to [3] which was also not cited), with respect to state-of-the-art robust pruning techniques for adversarially trained DNNs [4].

Kind regards.

[1] O. Özdenizci and R. Legenstein, "Training adversarially robust sparse networks via Bayesian connectivity sampling", ICML 2021.

[2] https://github.com/IGITUGraz/SparseAdversarialTraining

[3] Y. Fu, et al., "Drawing Robust Scratch Tickets: Subnetworks with Inborn Robustness Are Found within Randomly Initialized Networks", NeurIPS 2021.

[4] V. Sehwag, et al., "Hydra: Pruning adversarially robust neural networks.", NeurIPS 2020.

---

> ### Author Response · Authors · 2021-11-21
> **Response for Ozan Ozdenizci**
>
> Thanks for pointing out the references. We have cited all aforementioned publications with a detailed discussion in our revision. Meanwhile, we respectfully argue the difference between our work and existing literature and provide point-to-point feedback as follows.
>
> **[A. Compared to Özdenizci 2021.]**
>
> We clarify that our flying bird(+) is different from the approach in [Özdenizci 2021] at both levels of goal and methodologies.
>
> *<Different goals.>* [Özdenizci 2021] pursues a better adversarial robust testing accuracy for compressed networks. Our work aims to investigate the relationship between sparsity and robust generalization, and reveal that introducing appropriate sparsity (e.g., LTH-based static sparsity or dynamic sparsity) into adversarial training substantially alleviates the robust generalization gap and maintains comparable or even better standard/robust accuracies.
>
> *<Different methodologies.>* [Özdenizci 2021] samples network connectivity from a learned posterior to form a sparse subnetwork. However, our flying bird first removes the parameters with the lowest magnitude, which ensures a small term of the first-order Taylor approximation of the loss and thus limits the impact on the output of networks. And then it allows new connectivity with the largest gradient to grow for reducing the loss quickly.
>
> Moreover, our proposed flying bird+ not only learns the sparse topologies, but also is capable of adaptively adjusting the network capacity to determine the right parameterization level “on-demand” during training, while other sparse training methods (e.g., [Özdenizci 2021]) stick to a fixed parameter budget.
>
> Lastly, we would like to conduct extra experiments of [Özdenizci 2021] with the official implementation (https://github.com/IGITUGraz/SparseAdversarialTraining). For a fair comparison, we train a ResNet-18 on CIFAR-10 at 80% sparsity using PGD-10 for training, and then take PGD-20 for evaluation. Results show that the algorithm in [Özdenizci 2021] reaches 50.59% RA and remains a 35.87% robust generalization gap, while our flying bird/flying bird+ achieves 1.03%/1.11% higher RA and 6.97%/11.98% narrower robust generalization gaps.
>
> **[B. More Rigorous Evaluations.]**
>
> Thanks for the suggestions. Actually, we have already evaluated our approaches under improved attacks, i.e., Auto-Attack and CW-Attack. As shown in Table A7. our methods shrink the robust generalization gaps by up to 30.76% on CIFAR-10/100 with ResNet-18 at 80% sparsity. Furthermore, Table A6 demonstrates the maintained enhanced robustness under unseen transfer attacks, which excludes the possibility of gradient masking.
>
> **[C. Compared to Fu 2021 and Sehwag 2020.]**
>
> We want to point out that [Fu 2021] is publicly available on 26 Oct 2021 UTC on ArXiv, which is later than the ICLR2022 submission deadline. It is interesting to investigate it and we will leave it to our future works.
>
> For [Sehwag 2020], we respectfully argue that it can not be directly compared with our Robust Bird since they differ a lot in training schemes.
>
> *<Different starting points.>* [Sehwag 2020] starts from a robust pre-trained dense network, which requires at least hundreds of epochs for adversarial training. However, our robust bird’s pre-training only needs a few epochs of standard training. Therefore, [Sehwag 2020] has significantly heavier computational costs, compared to ours.
>
> *<Different objective functions.>* In [Sehwag 2020], the authors adopt TRADES [Zhang 2019] for adversarial training which also requires auxiliary inputs of clean images. However, our methods follow the classical adversarial training [Madry 2018] and only take adversarial perturbed samples as input.
>
> *<Extra training data or not?>* For CIFAR-10 experiments, [Sehwag 2020] uses 500k additional pseudo-labeled images from the TinyImages dataset with a robust semi-supervised training approach. However, all our methods and experiments do not leverage any external data.
>
> Lastly, we would like to conduct extra experiments of [Sehwag 2020] with the official implementation (https://github.com/inspire-group/hydra), under a unified fair setup. We replace the TRADES method with vanilla PGD-10 AT and train a ResNet-18 on CIFAR-10 at 80% sparsity. All evaluations are conducted under PGD-20. We observe that HYDRA obtains a 48.38% RA and 39.65% robust generalization gap, which is inferior to our flying bird/flying bird+ (3.24%/3.32% lower RA and 10.75%/15.76% larger robust generalization gaps).
>
> References.
>
> [Özdenizci 2021] Training adversarially robust sparse networks via Bayesian connectivity sampling.
>
> [Fu 2021] Drawing Robust Scratch Tickets: Subnetworks with Inborn Robustness Are Found within Randomly Initialized Networks.
>
> [Sehwag 2020] Hydra: Pruning adversarially robust neural networks.
>
> [Zhang 2019] Theoretically Principled Trade-off between Robustness and Accuracy.
>
> [Madry 2018] Towards deep learning models resistant to adversarial attacks.

---

### Decision · Program_Chairs · 2022-01-20

**Decision:**

Accept (Poster)

**Comment:**

This paper focuses on leveraging static and dynamic sparsity in efficient robust training. The proposed methods can significantly mitigate the robust generalization gap while retaining competitive performance (standard/robust accuracy) with substantially reduced computation budgets. The philosophy behind sounds quite interesting to me, namely, sparsity allevating overfitting and improving training efficiency simultaneously. This philosophy leads to two novel algorithms design, i.e., static Robust Bird training, and dynamic Flying Bird training.

The clarity and novelty are clearly above the bar of ICLR. While the reviewers had some concerns on the significance, the authors did a particularly good job in their rebuttal. Thus, most of us have agreed to accept this paper for publication! Please include the additional experimental results in the next version.